# Advancing Out-of-Distribution Detection via Local Neuroplasticity

**Alessandro Canevaro**[*,1,2]    **Julian Schmidt**[1]    **Mohammad Sajad Marvi**[1]
**Hang Yu**[1]    **Georg Martius**[2]    **Julian Jordan**[1]
[1]Mercedes-Benz AG, Sindelfingen, Germany    [2]University of Tübingen, Tübingen, Germany
[*] `alessandro.canevaro@mercedes-benz.com`

## Abstract

In the domain of machine learning, the assumption that training and test data share the same distribution is often violated in real-world scenarios, requiring effective out-of-distribution (OOD) detection. This paper presents a novel OOD detection method that leverages the unique local neuroplasticity property of Kolmogorov-Arnold Networks (KANs). Unlike traditional multilayer perceptrons, KANs exhibit local plasticity, allowing them to preserve learned information while adapting to new tasks. Our method compares the activation patterns of a trained KAN against its untrained counterpart to detect OOD samples. We validate our approach on benchmarks from image and medical domains, demonstrating superior performance and robustness compared to state-of-the-art techniques. These results underscore the potential of KANs in enhancing the reliability of machine learning systems in diverse environments.

## 1 Introduction

Most machine learning algorithms operate under the assumption that training and test data share the same distribution. However, this assumption frequently fails in real-world scenarios where models encounter out-of-distribution (OOD) data—samples that deviate from the training distribution, such as those belonging to novel categories. This mismatch can significantly impair a model's accuracy and reliability. As a result, OOD detection has become a critical area of research, aiming to discern when inputs fall outside the scope of the distribution used for training. Effective OOD detection not only enhances a model's robustness by identifying and handling these anomalous inputs but also ensures that the model maintains reliable performance on known, in-distribution data (Yang et al., 2022).

OOD detection poses a significant challenge due to the diverse nature of OOD types. While many OOD detectors excel when tested against specific OOD datasets, they often struggle to maintain high performance across a broad range of OOD samples. As stated by Zhang et al. (2023a) *there is no single winner that always outperforms others across multiple datasets*. One reason for this inconsistency is that OOD instances can vary widely, from subtle variations near the distribution boundary to completely dissimilar and far-off examples. As a result, developing a universal OOD detection method that performs robustly across multiple datasets, spanning near to far OOD samples, remains challenging.

In this paper, we present a novel OOD detection method using Kolmogorov-Arnold Networks (KANs) (Liu et al., 2024b). KANs are neural networks with a unique architecture that enhances interpretability, improves the accuracy-to-parameter ratio, and mitigates catastrophic forgetting compared to multilayer perceptrons (MLPs). Our approach takes advantage of KANs' distinctive property of local neuroplasticity—a characteristic absent in traditional MLPs due to their reliance on shared, non-trainable activation functions. In contrast, KANs demonstrate local plasticity owing to their spline-based architecture. This characteristic ensures that learning a new task impacts only the regions of the network activated by the training data, thereby preserving the integrity of distant and unrelated regions.

As illustrated in Figure 1, our method compares the activation patterns of two identically initialized KANs: one trained on In-Distribution (InD) data and the other left untrained. OOD samples will predominantly trigger the regions of the trained network that were not adapted during the learning phase, thus the samples will produce a response closer to the untrained network.

Conversely, InD samples will mostly engage the neurons that have been trained, resulting in a noticeable difference in the activation between the two models.

We tested our method on seven different benchmarks from two different domains: the OpenOOD CIFAR-10, CIFAR-100, ImageNet-200 full-spectrum (FS), and ImageNet-1K FS (Yang et al., 2022) for the image domain, and the Ethnicity, Age, and Synthetic OOD benchmarks for the tabular medical data domain (Azizmalayeri et al., 2023). Our experiments demonstrate that the KAN detector outperforms current State-Of-The-Art (SOTA) techniques across all seven benchmarks on the overall average AUROC that considers both near and far OOD. Additionally, in contrast to many other SOTA methods, our approach's performance does not vary significantly based on the number of training samples. This indicates that leveraging KANs leads to highly effective OOD detection, underscoring the potential of this novel architecture in developing more robust machine learning systems capable of operating reliably in diverse and unpredictable environments.

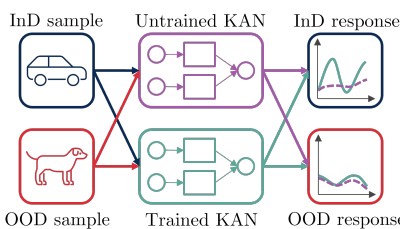

Figure 1: Overview of the proposed method: the detector compares the activation function response of a trained KAN model with its untrained counterpart. A difference in the response indicates the sample is InD, a similar response suggests it is OOD.

## 2 KAN-BASED OOD DETECTION

This section begins by providing a short background on KANs and their working principle. Next, we delve into the core concept underlying our proposed method for OOD detection. Finally, we describe the primary limitation of the KAN detector and propose a strategy to enable its deployment in complex real-world scenarios.

### 2.1 BACKGROUND

KANs are neural network architectures based on the Kolmogorov-Arnold representation theorem. This theorem states that any continuous multivariate function can be represented as a sum of continuous functions of a single variable. Hence, KANs approximate high-dimensional functions using simpler, univariate components, effectively addressing the curse of dimensionality in machine learning.

In practice, KANs decompose multivariate functions into univariate B-spline functions with learnable coefficients. Let $x_p$ be the p-th component (feature) of the input vector $\mathbf{x} \in \mathbb{R}^{n_{\text{in}}}$ and let $y_q$ be the q-th component (feature) of the output vector $\mathbf{y} \in \mathbb{R}^{n_{\text{out}}}$. A KAN layer transforms $\mathbf{x}$ into $\mathbf{y}$ using a matrix of univariate functions $\Phi = \{\phi_{p,q}\}$, where each $\phi_{p,q}$ is parameterized by a B-spline. Each B-spline consists of a linear combination of $G + k$ B-spline basis functions with learnable coefficients $c_{p,q,i}$. The spline order is denoted as $k$ (usually $k = 3$) and $G$ is the grid size.

$$y_q = \sum_p \phi_{p,q}(x_p) \quad \text{with:} \quad \phi_{p,q}(x_p) = \sum_{i=0}^{G+k} c_{p,q,i} B_i(x_p). \quad (1)$$

KAN layers can be stacked to construct deeper networks, allowing for complex transformations across multiple stages. Performance is further enhanced by incorporating residual connections, which add flexibility to the spline functions through trainable weights and additional basis functions (Liu et al., 2024b).

Local neuroplasticity in KANs is facilitated by two key properties. First, each input feature $x_p$ is processed independently by its own set of activation functions $\{\phi_{p,q} \mid \forall q\}$. Second, during

backpropagation, only the spline coefficients near sample $x_p$ are modified, leaving the other areas of the activation function largely unchanged.

## 2.2 OOD Detection with KANs

We propose leveraging the local plasticity of KANs for OOD detection. The InD data seen during training only affects specific regions (spline grid coefficients) of the network. By determining whether a region contains InD data and inspecting which regions are activated by each sample, the KAN-based detector can distinguish between InD and OOD samples. This differentiation is achieved by comparing the output of the trained activation functions with their values prior to training. The step-by-step procedure is as follows:

- **Setup**: Initialize an untrained KAN and create a copy. Train one KAN with the InD dataset while keeping the other untrained.
- **Detection**: Perform a forward pass on both networks with the given sample **x**, and save the output of the activation functions:

$$\phi_{p,q}^{\text{trained}}(x_p), \quad \phi_{p,q}^{\text{untrained}}(x_p) \quad \forall p, q \tag{2}$$

Compute the difference between the responses:

$$\Delta_{p,q}(x_p) = \left| \phi_{p,q}^{\text{trained}}(x_p) - \phi_{p,q}^{\text{untrained}}(x_p) \right|. \tag{3}$$

Analyze the difference matrix $\Delta$. OOD samples will tend to have a higher ratio of the entries in the $\Delta$ matrix close to zero. To obtain a scalar InD score $S(\mathbf{x})$, we aggregate the differences using a scoring function $F_{\text{score}}$ (detailed in Appendix A.1):

$$S(\mathbf{x}) = F_{\text{score}}(\Delta(\mathbf{x})). \tag{4}$$

To clarify our method's working principle, let us rewrite $\Delta_{p,q}(x_p)$ using Eq. 1:

$$\Delta_{p,q}(x_p) = \sum_i \left| c_{p,q,i}^{\text{trained}} - c_{p,q,i}^{\text{untrained}} \right| \cdot B_i(x_p). \tag{5}$$

The terms $\left| c_{p,q,i}^{\text{trained}} - c_{p,q,i}^{\text{untrained}} \right|$ define the locations within the network where InD information is stored, while $B_i(x_p)$ serves as a mask and specify the regions activated by the sample **x**. Consequently, multiplying these two terms provides a quantitative measure of the overlap between InD regions and the given sample. This overlap is subsequently utilized to compute the InD score.

Once the InD score is obtained, it is thresholded to classify the sample as InD or OOD:

$$D(\mathbf{x}) = \begin{cases} \text{InD}, & \text{if } S(\mathbf{x}) \geq \lambda \\ \text{OOD}, & \text{if } S(\mathbf{x}) < \lambda, \end{cases} \tag{6}$$

where $\lambda$ is a predefined threshold. A test sample with a InD score less than $\lambda$ is categorized as OOD. Otherwise, it is classified as InD.

Figure 2 illustrates the working principle of the proposed algorithm using a modified version of the toy example proposed by Liu et al. (2024b). The dataset is a one-dimensional regression task featuring five Gaussian peaks. We used two of these peaks as the training set and InD test set, while the remaining three peaks are the OOD test set. Here the KAN model is composed of a single layer with one input and one output, i.e. a single univariate function $\phi$ with 200 spline coefficients.

## 2.3 Capturing the joint feature distribution

Like MLPs, KANs are capable of processing multivariate inputs and producing multivariate outputs. However, differently from MLPs where activation functions receive a weighted sum of all inputs, in KANs, each activation function receives only a single input. While this characteristic allows the KAN detector to effectively capture the marginal distributions of input features, it also constrains its ability to model the joint distribution of features.

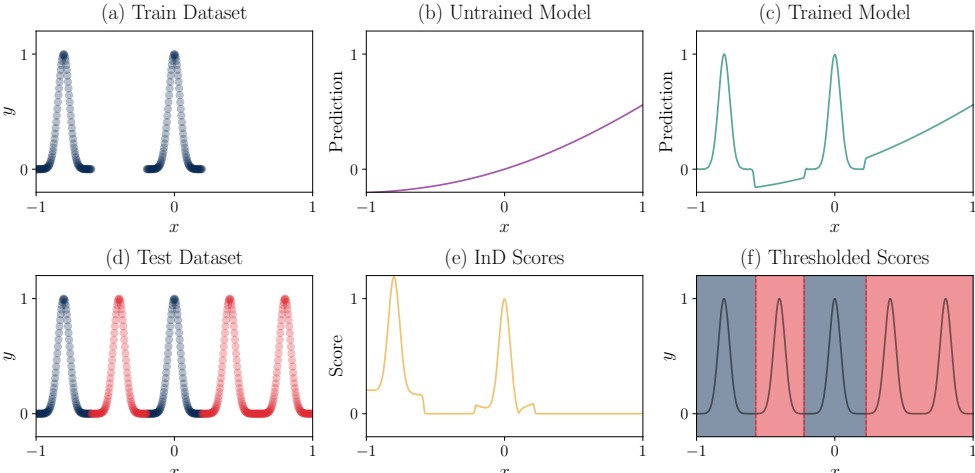

Figure 2: (a) Visualization of the training dataset, showing the relationship between inputs and targets. (b) Response of the untrained KAN model across the entire input range. (c) Response of the trained KAN model across the entire input range. (d) Test dataset illustrating inputs versus targets, created by combining the training dataset (InD) with three additional Gaussian peaks over the remaining input range (OOD). (e) InD score $S(\mathbf{x})\forall \mathbf{x} \in [-1, 1]$ using the median as scoring function ($F_{\text{score}}$). (f) Final results after applying a threshold ($\lambda = 1e - 3$) to the InD scores: blue regions indicate predicted InD areas and red regions indicate predicted OOD areas.

To overcome this limitation we propose to partition the InD dataset and train separate KAN models for each partition. In this way, the complex training distribution is decomposed into smaller parts that can be accurately described using only the marginal feature distribution. Various techniques can be employed to partition the dataset. A simple, yet effective approach is to split the dataset based on class labels. An alternative approach, which also works when class labels are absent, such as in regression tasks, is to apply a clustering algorithm like k-means (Lloyd, 1982). Formally, the dataset $\mathcal{D}$ is partitioned into $\mathcal{P}$ non-overlapping subsets $\mathcal{D}_1, \mathcal{D}_2, \ldots, \mathcal{D}_{\mathcal{P}}$. For each partition $\mathcal{D}_i$, we train a separate detector, denoted as $\text{KAN}_i$. While the partition $\mathcal{D}_i$ is different for each $\text{KAN}_i$, the training task is always the same (e.g., classification). During inference, we compute the InD score for a sample $\mathbf{x}$ by aggregating the InD score from each KAN model. Let $\Delta^i(\mathbf{x})$ be the difference matrix of $\text{KAN}_i$:

$$S(\mathbf{x}) = F_{\text{agg.}}\big(F_{\text{score}}(\Delta^1(\mathbf{x})), F_{\text{score}}(\Delta^2(\mathbf{x})), \ldots, F_{\text{score}}(\Delta^{\mathcal{P}}(\mathbf{x})\big), \tag{7}$$

where $F_{\text{agg.}}$ is a suitable aggregation function, such as the maximum function. Since the partitions are non-overlapping, for InD samples, there will be only one model that recognises the sample as InD (high InD score), while the other will flag it as OOD (low InD score).

Through this partitioning, our detector is now composed of multiple KAN models, and the strategy resembles ensemble methods. Furthermore, if all models are initialized with the same weights, the untrained KAN can be shared, reducing the number of forward passes at inference time.

To demonstrate the effectiveness of our proposed improvement method, we designed a specialized L-shaped dataset where the base KAN detector fails. This dataset consists of 2D points, with the training task being regression to a predefined constant. Figure 3 illustrates the results, showing the performance of the default KAN detector compared to the partitioning method.

## 3 EXPERIMENTS

First, we describe the benchmarks, metrics, and implementation details used in our study. The results demonstrate the superior performance of our method, highlighting its key advantages. Finally, a comprehensive ablation study analyzes each hyperparameter and component, elucidating their impact on performance.

(a) Default approach

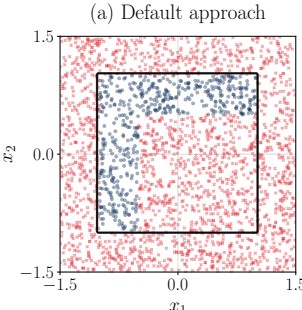

(b) Dataset partitioning

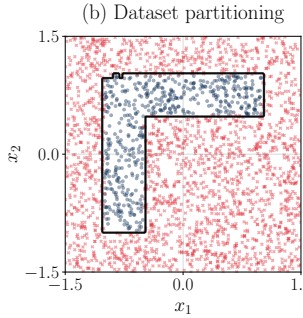

Figure 3: 2D L-shape toy dataset: The blue point cloud shows the training distribution and the red points are OOD test samples. The black contour represents the thresholded score function (median) separating InD from OOD samples. (a) Default KAN detector limited by the marginal distribution of $x_1$ and $x_2$. (b) Improved performance by partitioning the training dataset with KMeans clustering ($\mathcal{P} = 2$) and $F_{\text{agg.}} = \max$ as aggregation function.

## 3.1 EVALUATION PROTOCOL

**Setup.** The evaluation of the proposed method is performed on seven different benchmarks from two different domains: OOD detection in images and tabular medical data.

For OOD detection in images, the experimental setup adheres to the OpenOOD (Yang et al., 2022) benchmark protocol. We evaluate the KAN detector on the CIFAR-10 benchmark, using CIFAR-10 (Krizhevsky et al., b) as the InD dataset. The OOD datasets are categorized into near OOD datasets (CIFAR-100 (Krizhevsky et al., a) and Tiny ImageNet (TIN) (Le & Yang, 2015)) and far OOD datasets (MNIST (Deng, 2012), SVHN (Netzer et al., 2011), Textures (Cimpoi et al., 2014), and Places365 (Zhou et al., 2018)). The CIFAR-100 benchmark contains the same datasets as the CIFAR-10 benchmark except for the CIFAR-10 and CIFAR-100 datasets which have an inverted role (CIFAR-100 as training data and CIFAR-10 as OOD dataset). To evaluate the scalability of our method, we also tested it on the ImageNet-200 FS and ImageNet-1K FS benchmarks. Compared to CIFAR-10 and CIFAR-100, these benchmarks features five to twenty times more training images, each with a size seven times larger. The full-spectrum version increases the detection challenge and, at the same time, makes it closer to real-world applications by enriching the InD test set with covariate-shifted InD samples (Yang et al., 2023). The datasets used in this benchmark are: ImageNet-200 or ImageNet-1K (Deng et al., 2009) as training set, ImageNet-V2 (Recht et al., 2019), ImageNet-C (Hendrycks & Dietterich, 2019), ImageNet-R (Hendrycks et al., 2021) as covariate-shifted InD test set, SSB-hard (Vaze et al., 2022), NINCO (Bitterwolf et al., 2023) as near OOD, and iNaturalist (Van Horn et al., 2018), Textures, OpenImage-O (Wang et al., 2022) as far OOD.

For OOD detection in tabular medical data, we follow the benchmark proposed by Azizmalayeri et al. (2023). We consider the benchmarks derived from the eICU dataset (Pollard et al., 2018), which contains clinical data of tens of thousands of Intensive Care Unit (ICU) patients in several hospitals. In the near OOD benchmarks, the eICU dataset is divided into InD and OOD according to some features such as ethnicity (Caucasian as InD) or age (older than 70 as InD). The feature used for splitting the dataset is then removed. In the synthetic OOD benchmark, the OOD data is generated by scaling a single feature from the InD set by a factor $\mathcal{F}$. For each factor, the experiment is repeated 100 times with different features, to minimize the impact of the chosen feature. By varying the scaling factor, the generated samples range from near to far OOD.

In contrast to training-time regularization methods (e.g., MOS (Huang & Li, 2021), CIDER (Ming et al., 2023)), our detector operates in a post-hoc manner and can be seamlessly integrated with any pre-trained classifier, regardless of model architecture, training procedures, or types of OOD data. The backbone is used to perform the classification or regression task and in the case of post-hoc methods it is trained independently from the detector. The OOD detector only uses the latent features of the backbone for InD/OOD classification. The considered OpenOOD benchmarks employ a pre-trained ResNet backbone (He et al., 2015) for feature extraction, while the tabular medical benchmarks use an FT-Transformer backbone (Gorishniy et al., 2021).

Given that the benchmarks we considered are all based on classification tasks and require a pre-trained backbone network, we conducted additional experiments on regression-based datasets, applying the detector directly to the data without a feature extractor. The results, presented in Appendix A.2, demonstrate that our method also performs well in these scenarios.

**Metrics.** In all benchmarks, the primary metric used to evaluate the OOD detection performance is the Area Under the Receiver Operating Characteristic curve (AUROC). This threshold-free metric provides a robust assessment of the model's ability to distinguish between InD and OOD samples.

In our evaluation, we focus on the average AUROC across all test datasets, including both near and far OOD (for more details on how the overall average is computed see Appendix A.3). This approach is motivated by the desire to develop a method that performs well across diverse datasets, as real-world applications often encounter unknown types of OOD samples. We acknowledge that achieving consistent performance across multiple datasets is challenging, as many methods excel on specific datasets but struggle to generalize.

Following the OpenOOD benchmark guidelines, we report the results averaged over three seeds, corresponding to three pre-trained backbones. This does not apply to the ImageNet-1K FS benchmark where only one pre-trained backbone is available. The results are averaged over five seeds for the tabular medical data benchmarks. Our approach also introduces some stochasticity due to the KAN initialization. To assess its impact on performance, we initialized the detector with five different seeds for each pre-trained backbone. The results indicate that the stochasticity due to the KAN initialization is lower than the one due to the backbone training (see Appendix A.4).

**Implementation details.** The detectors are trained using the InD dataset. During the evaluation phase, InD scores are calculated for all the test data. All hyperparameters are tuned using the validation set, according to the OpenOOD benchmark guidelines. The tabular medical data benchmarks follow a similar structure.

On all benchmarks, we used the median as the scoring function and the maximum as the aggregation function. The median is particularly effective due to its robustness to outliers, making it reliable for distinguishing between InD and OOD samples, as illustrated in Figure 4. These choices for both (scoring and aggregation) are further motivated in Appendix A.1.

The latent features of the backbones exhibited a highly skewed distribution. To address this skewness and achieve a more balanced distribution that fully utilizes the KAN's grid range, we applied histogram normalization.

We leverage information from multiple latent layers of the pre-trained backbone. As demonstrated by Liu et al.

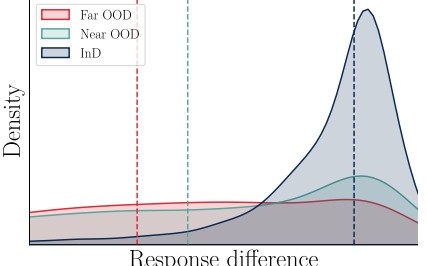

Figure 4: Distribution of activation's differences ($\Delta$) for three different samples (InD, near and far OOD). The InD sample tends to produce bigger values in the $\Delta$ matrix compared to the OOD samples. Using the median as a scoring function (vertical dashed lines) effectively separates InD from OOD.

(2024a), this multi-layer integration enriches the feature representation, leading to improved detection accuracy. Specifically, the authors claim that the last layer contains predominantly semantic information while including the layers closer to the input allows the detector to capture also the covariate information.

## 3.2 RESULTS

Table 1 presents the results of our experiments on the CIFAR-10 and CIFAR-100 benchmarks, comparing the KAN detector with several SOTA OOD detection methods (see Appendix A.5 for a list of all the considered baselines). On top of the numerous baselines provided by the benchmark we also compare our approach to the current best post-hoc method on the CIFAR leaderboard: the NAC (Liu et al., 2024a). The results show that the KAN detector outperforms all previous methods on both benchmarks, demonstrating the effectiveness of leveraging spline-based local activation functions for OOD detection. In each column of this and the following tables, we highlight in **bold** the best-performing method. Where multiple seeds of the backbone are available we also highlight

any other methods that do not show a statistically significant difference from the best-performing method. Statistical significance is assessed using Welch's t-test with $p < 0.05$.

Table 1: Comparison of OOD detection performance (AUROC) on CIFAR-10 and CIFAR-100 benchmarks.

| Method | Near OOD | | Far OOD | | | | Avg Near | Avg Far | Avg Overall |
|---|---|---|---|---|---|---|---|---|---|
| | **CIFAR** | **TIN** | **MNIST** | **SVHN** | **Textures** | **Places365** | | | |
| | | | | **CIFAR-10 Benchmark** | | | | | |
| OpenMax | $86.91_{\pm0.31}$ | $88.32_{\pm0.28}$ | $90.50_{\pm0.44}$ | $89.77_{\pm0.45}$ | $89.58_{\pm0.60}$ | $88.63_{\pm0.28}$ | $87.62_{\pm0.29}$ | $89.62_{\pm0.19}$ | $88.95_{\pm0.41}$ |
| ODIN | $82.18_{\pm1.87}$ | $83.55_{\pm1.84}$ | $\mathbf{95.24}_{\pm1.96}$ | $84.58_{\pm0.77}$ | $86.94_{\pm2.26}$ | $85.07_{\pm1.24}$ | $82.87_{\pm1.85}$ | $87.96_{\pm0.61}$ | $86.26_{\pm1.73}$ |
| MDS | $83.59_{\pm2.27}$ | $84.81_{\pm2.53}$ | $90.10_{\pm2.41}$ | $91.18_{\pm0.47}$ | $92.69_{\pm1.06}$ | $84.90_{\pm2.54}$ | $84.20_{\pm2.40}$ | $89.72_{\pm1.36}$ | $87.88_{\pm2.05}$ |
| MDSEns | $61.29_{\pm0.23}$ | $59.57_{\pm0.53}$ | $\mathbf{99.17}_{\pm0.41}$ | $66.56_{\pm0.58}$ | $77.40_{\pm0.28}$ | $52.47_{\pm0.15}$ | $60.43_{\pm0.26}$ | $73.90_{\pm0.27}$ | $69.41_{\pm0.40}$ |
| RMDS | $88.83_{\pm0.35}$ | $90.76_{\pm0.27}$ | $93.22_{\pm0.80}$ | $91.84_{\pm0.26}$ | $92.23_{\pm0.23}$ | $\mathbf{91.51}_{\pm0.11}$ | $89.80_{\pm0.28}$ | $92.20_{\pm0.21}$ | $91.40_{\pm0.40}$ |
| Gram | $58.33_{\pm4.49}$ | $58.98_{\pm5.19}$ | $72.64_{\pm2.34}$ | $\mathbf{91.52}_{\pm4.45}$ | $62.34_{\pm8.27}$ | $60.44_{\pm3.41}$ | $58.66_{\pm4.83}$ | $71.73_{\pm3.20}$ | $67.37_{\pm5.04}$ |
| ReAct | $85.93_{\pm0.83}$ | $88.29_{\pm0.44}$ | $\mathbf{92.81}_{\pm3.03}$ | $89.12_{\pm3.19}$ | $89.38_{\pm1.49}$ | $\mathbf{90.35}_{\pm0.78}$ | $87.11_{\pm0.61}$ | $90.42_{\pm1.41}$ | $89.31_{\pm1.96}$ |
| VIM | $87.75_{\pm0.28}$ | $89.62_{\pm0.33}$ | $94.76_{\pm0.38}$ | $94.50_{\pm0.48}$ | $\mathbf{95.15}_{\pm0.34}$ | $89.49_{\pm0.39}$ | $88.68_{\pm0.28}$ | $93.48_{\pm0.24}$ | $91.88_{\pm0.37}$ |
| KNN | $\mathbf{89.73}_{\pm0.14}$ | $\mathbf{91.56}_{\pm0.26}$ | $94.26_{\pm0.38}$ | $92.67_{\pm0.30}$ | $93.16_{\pm0.24}$ | $\mathbf{91.77}_{\pm0.23}$ | $90.64_{\pm0.20}$ | $92.96_{\pm0.14}$ | $92.19_{\pm0.27}$ |
| ASH | $74.11_{\pm1.55}$ | $76.44_{\pm0.61}$ | $83.16_{\pm4.66}$ | $73.46_{\pm6.41}$ | $77.45_{\pm2.39}$ | $79.89_{\pm3.69}$ | $75.27_{\pm1.04}$ | $78.49_{\pm2.58}$ | $77.42_{\pm3.76}$ |
| SHE | $80.31_{\pm0.69}$ | $82.76_{\pm0.43}$ | $90.43_{\pm4.76}$ | $86.38_{\pm1.32}$ | $81.57_{\pm1.21}$ | $82.89_{\pm1.22}$ | $81.54_{\pm0.51}$ | $85.32_{\pm1.43}$ | $84.06_{\pm2.16}$ |
| GEN | $87.21_{\pm0.36}$ | $89.20_{\pm0.25}$ | $93.83_{\pm2.14}$ | $91.97_{\pm0.66}$ | $90.14_{\pm0.76}$ | $89.46_{\pm0.65}$ | $88.20_{\pm0.30}$ | $91.35_{\pm0.69}$ | $90.30_{\pm1.02}$ |
| NAC | $\mathbf{89.83}_{\pm0.29}$ | $\mathbf{92.02}_{\pm0.20}$ | $94.86_{\pm1.37}$ | $96.06_{\pm0.47}$ | $\mathbf{95.64}_{\pm0.45}$ | $\mathbf{91.85}_{\pm0.28}$ | $90.93_{\pm0.23}$ | $94.60_{\pm0.50}$ | $\mathbf{93.37}_{\pm0.64}$ |
| **KAN** | $\mathbf{90.06}_{\pm0.47}$ | $91.92_{\pm0.52}$ | $\mathbf{97.86}_{\pm0.73}$ | $\mathbf{97.39}_{\pm0.42}$ | $\mathbf{95.85}_{\pm0.28}$ | $91.64_{\pm0.91}$ | $\mathbf{90.99}_{\pm0.50}$ | $\mathbf{95.69}_{\pm0.22}$ | $\mathbf{94.12}_{\pm0.50}$ |
| | | | | **CIFAR-100 Benchmark** | | | | | |
| OpenMax | $74.38_{\pm0.37}$ | $78.44_{\pm0.14}$ | $76.01_{\pm1.39}$ | $82.07_{\pm1.53}$ | $80.56_{\pm0.09}$ | $79.29_{\pm0.40}$ | $76.41_{\pm0.25}$ | $79.48_{\pm0.41}$ | $78.46_{\pm0.88}$ |
| ODIN | $78.18_{\pm0.14}$ | $81.63_{\pm0.08}$ | $83.79_{\pm1.31}$ | $74.54_{\pm0.76}$ | $79.33_{\pm1.08}$ | $79.45_{\pm0.26}$ | $79.90_{\pm0.11}$ | $79.28_{\pm0.21}$ | $\mathbf{79.49}_{\pm0.77}$ |
| MDS | $55.87_{\pm0.22}$ | $61.50_{\pm0.28}$ | $67.47_{\pm0.81}$ | $70.68_{\pm6.40}$ | $76.26_{\pm0.69}$ | $63.15_{\pm0.49}$ | $58.69_{\pm0.09}$ | $69.39_{\pm1.39}$ | $65.82_{\pm2.66}$ |
| MDSEns | $43.85_{\pm0.31}$ | $48.78_{\pm0.19}$ | $\mathbf{98.21}_{\pm0.78}$ | $53.76_{\pm1.63}$ | $69.75_{\pm1.14}$ | $42.27_{\pm0.73}$ | $46.31_{\pm0.24}$ | $66.00_{\pm0.69}$ | $59.44_{\pm0.93}$ |
| RMDS | $77.75_{\pm0.19}$ | $82.55_{\pm0.02}$ | $79.74_{\pm2.49}$ | $84.89_{\pm1.10}$ | $83.65_{\pm0.51}$ | $\mathbf{83.40}_{\pm0.46}$ | $80.15_{\pm0.11}$ | $82.92_{\pm0.42}$ | $\mathbf{82.00}_{\pm1.15}$ |
| Gram | $49.41_{\pm0.58}$ | $53.91_{\pm1.58}$ | $80.71_{\pm4.15}$ | $\mathbf{95.55}_{\pm0.60}$ | $70.79_{\pm1.32}$ | $46.38_{\pm1.21}$ | $51.66_{\pm0.77}$ | $73.36_{\pm1.08}$ | $66.12_{\pm1.98}$ |
| ReAct | $78.65_{\pm0.05}$ | $82.88_{\pm0.08}$ | $78.37_{\pm1.59}$ | $83.01_{\pm0.97}$ | $80.15_{\pm0.46}$ | $80.03_{\pm0.11}$ | $80.77_{\pm0.05}$ | $80.39_{\pm0.49}$ | $\mathbf{80.52}_{\pm0.79}$ |
| VIM | $72.21_{\pm0.41}$ | $77.76_{\pm0.16}$ | $81.89_{\pm1.02}$ | $83.14_{\pm3.71}$ | $85.91_{\pm0.78}$ | $75.85_{\pm0.37}$ | $74.98_{\pm0.13}$ | $81.70_{\pm0.62}$ | $\mathbf{79.46}_{\pm1.62}$ |
| KNN | $77.02_{\pm0.25}$ | $\mathbf{83.34}_{\pm0.16}$ | $82.36_{\pm1.52}$ | $84.15_{\pm1.09}$ | $83.66_{\pm0.83}$ | $79.43_{\pm0.47}$ | $80.18_{\pm0.15}$ | $82.40_{\pm0.17}$ | $\mathbf{81.66}_{\pm0.87}$ |
| ASH | $76.48_{\pm0.30}$ | $79.92_{\pm0.20}$ | $77.23_{\pm0.46}$ | $85.60_{\pm1.40}$ | $80.72_{\pm0.70}$ | $78.76_{\pm0.16}$ | $78.20_{\pm0.15}$ | $80.58_{\pm0.66}$ | $\mathbf{79.79}_{\pm0.69}$ |
| SHE | $78.15_{\pm0.03}$ | $79.74_{\pm0.36}$ | $76.76_{\pm1.07}$ | $80.97_{\pm3.98}$ | $73.64_{\pm1.28}$ | $76.30_{\pm0.51}$ | $78.95_{\pm0.18}$ | $76.92_{\pm1.16}$ | $77.59_{\pm1.78}$ |
| GEN | $\mathbf{79.38}_{\pm0.04}$ | $\mathbf{83.25}_{\pm0.13}$ | $78.29_{\pm2.05}$ | $81.41_{\pm1.50}$ | $78.74_{\pm0.81}$ | $80.28_{\pm0.27}$ | $\mathbf{81.31}_{\pm0.08}$ | $79.68_{\pm0.75}$ | $\mathbf{80.23}_{\pm1.10}$ |
| NAC | $72.02_{\pm0.69}$ | $79.86_{\pm0.23}$ | $93.26_{\pm1.34}$ | $92.60_{\pm1.14}$ | $\mathbf{89.36}_{\pm0.54}$ | $73.06_{\pm0.63}$ | $75.94_{\pm0.41}$ | $\mathbf{87.07}_{\pm0.30}$ | $\mathbf{83.36}_{\pm0.84}$ |
| **KAN** | $72.97_{\pm0.17}$ | $81.37_{\pm0.22}$ | $92.29_{\pm1.85}$ | $87.16_{\pm4.46}$ | $\mathbf{89.43}_{\pm0.39}$ | $77.42_{\pm0.35}$ | $77.17_{\pm0.17}$ | $86.57_{\pm0.70}$ | $\mathbf{83.44}_{\pm1.99}$ |

The KAN detector ranks first on both ImageNet-200 FS and ImageNet-1K FS benchmarks as shown in Table 2 and it consistently ranks in the top three across all tabular medical data benchmarks as reported in Tables 3, 4 and 5.

In Appendix A.6 we report the results with all the baselines available in the benchmarks for the AUROC and FPR@95 metrics.

Moreover, our method demonstrates significant robustness to variations in the number of training samples. Table 6 analyses this phenomenon on the CIFAR-10 and CIFAR-100 benchmarks, comparing the performance of the three previously best-performing methods and our approach by evaluating all methods with different dataset sizes only. Unlike other methods that achieve peak performance only with an optimal number of training samples, our approach consistently performs well across different dataset sizes. The performance of VIM and KNN is closely tied to the size of the InD dataset, while NAC achieves its best results when only 2% of the training samples are used. In contrast, the KAN detector maintains high performance across a wide range of training dataset sizes, with only a minor decrease observed in the extreme case of five samples per class. Robustness to variations in training dataset sizes is crucial in real-world scenarios where the number of training samples may be insufficient to capture the underlying distribution's characteristics. Additionally, this property is advantageous when scaling to large datasets. We attribute the strong performance of our approach across all considered benchmarks also to this key characteristic.

## 3.3 Ablation Study

**Parameter analysis.** The main hyperparameters that regulate the performance of the proposed method are the number of partitions $\mathcal{P}$ and the grid size $G$. Table 7 illustrates the variations in AUROC performance as a function of the number of partitions obtained through k-means clustering. Increasing $\mathcal{P}$ enhances the detector's ability to capture the joint distribution of features, resulting

Table 2: Comparison of OOD detection performance (AUROC) on ImageNet-200 FS and ImageNet-1K FS benchmarks.

| Method | Near OOD | | Far OOD | | | Avg Near | Avg Far | Avg Overall |
|---|---|---|---|---|---|---|---|---|
| | SSB-hard | NINCO | iNaturalist | Textures | OpenImage-O | | | |
| | | | **ImageNet-200 FS Benchmark** | | | | | |
| OpenMax | 47.64$_{\pm0.20}$ | 54.15$_{\pm0.23}$ | 72.44$_{\pm0.87}$ | 69.12$_{\pm0.36}$ | 62.31$_{\pm0.24}$ | 50.89$_{\pm0.18}$ | 67.96$_{\pm0.39}$ | 61.13$_{\pm0.46}$ |
| ODIN | 44.31$_{\pm0.02}$ | 52.36$_{\pm0.08}$ | 70.19$_{\pm0.92}$ | 67.10$_{\pm0.34}$ | 61.48$_{\pm0.31}$ | 48.33$_{\pm0.05}$ | 66.25$_{\pm0.42}$ | 59.09$_{\pm0.46}$ |
| MDS | 48.59$_{\pm0.88}$ | 56.65$_{\pm0.94}$ | 68.25$_{\pm1.51}$ | 73.84$_{\pm0.75}$ | 61.90$_{\pm0.57}$ | 52.62$_{\pm0.90}$ | 68.00$_{\pm0.87}$ | 61.85$_{\pm0.98}$ |
| MDSEns | 34.22$_{\pm0.44}$ | 41.58$_{\pm0.17}$ | 43.63$_{\pm0.48}$ | 67.54$_{\pm0.35}$ | 48.38$_{\pm0.36}$ | 37.90$_{\pm0.20}$ | 53.18$_{\pm0.39}$ | 47.07$_{\pm0.38}$ |
| RMDS | 56.24$_{\pm0.62}$ | 60.95$_{\pm0.94}$ | 71.71$_{\pm1.49}$ | 64.61$_{\pm1.07}$ | 63.52$_{\pm0.83}$ | 58.59$_{\pm0.77}$ | 66.62$_{\pm1.11}$ | 63.41$_{\pm1.03}$ |
| Gram | **59.12**$_{\pm0.73}$ | **63.35**$_{\pm0.76}$ | 58.42$_{\pm0.75}$ | 75.86$_{\pm0.10}$ | 61.51$_{\pm0.39}$ | **61.23**$_{\pm0.74}$ | 65.26$_{\pm0.31}$ | 63.65$_{\pm0.61}$ |
| ReAct | 47.25$_{\pm0.57}$ | 53.84$_{\pm0.55}$ | 69.45$_{\pm3.94}$ | 71.45$_{\pm2.04}$ | 62.30$_{\pm2.32}$ | 50.55$_{\pm0.19}$ | 67.73$_{\pm2.76}$ | 60.86$_{\pm2.27}$ |
| VIM | 45.34$_{\pm0.72}$ | 57.09$_{\pm1.03}$ | 71.34$_{\pm1.68}$ | **82.54**$_{\pm0.73}$ | 65.70$_{\pm0.94}$ | 51.22$_{\pm0.86}$ | 73.19$_{\pm1.10}$ | 64.40$_{\pm1.08}$ |
| KNN | 44.05$_{\pm0.42}$ | 54.51$_{\pm0.62}$ | 71.53$_{\pm1.32}$ | 81.88$_{\pm0.19}$ | 62.12$_{\pm0.79}$ | 49.28$_{\pm0.51}$ | 71.84$_{\pm0.72}$ | 62.82$_{\pm0.77}$ |
| ASH | 50.96$_{\pm0.93}$ | 58.51$_{\pm0.60}$ | 77.96$_{\pm1.58}$ | 79.39$_{\pm0.61}$ | **69.09**$_{\pm0.71}$ | 54.74$_{\pm0.74}$ | 75.48$_{\pm0.95}$ | 67.18$_{\pm0.96}$ |
| SHE | 52.82$_{\pm0.65}$ | 56.64$_{\pm0.69}$ | 72.20$_{\pm2.65}$ | 74.27$_{\pm0.63}$ | 64.95$_{\pm1.25}$ | 54.73$_{\pm0.67}$ | 70.47$_{\pm1.39}$ | 64.18$_{\pm1.41}$ |
| GEN | 48.33$_{\pm0.27}$ | 54.85$_{\pm0.42}$ | 68.94$_{\pm0.63}$ | 66.58$_{\pm0.47}$ | 60.87$_{\pm0.28}$ | 51.59$_{\pm0.34}$ | 65.46$_{\pm0.44}$ | 59.91$_{\pm0.43}$ |
| NAC | 45.42$_{\pm0.11}$ | 53.80$_{\pm0.08}$ | 65.83$_{\pm1.22}$ | 74.41$_{\pm0.35}$ | 60.79$_{\pm0.23}$ | 49.61$_{\pm0.06}$ | 67.01$_{\pm0.53}$ | 60.05$_{\pm0.58}$ |
| **KAN** | **58.37**$_{\pm0.47}$ | 61.10$_{\pm0.53}$ | **84.13**$_{\pm0.35}$ | 83.30$_{\pm0.35}$ | **70.40**$_{\pm0.26}$ | 59.74$_{\pm0.46}$ | **79.28**$_{\pm0.18}$ | **71.46**$_{\pm0.40}$ |
| | | | **ImageNet-1K FS Benchmark** | | | | | |
| OpenMax | 53.79 | 60.28 | 80.30 | 73.54 | 71.88 | 57.03 | 75.24 | 67.96 |
| ODIN | 54.22 | 60.59 | 77.43 | 76.04 | 73.40 | 57.41 | 75.62 | 68.34 |
| MDS | 39.22 | 52.83 | 54.06 | 86.26 | 60.75 | 46.02 | 67.02 | 58.62 |
| MDSEns | 37.13 | 47.80 | 53.32 | 73.39 | 53.24 | 42.47 | 59.98 | 52.98 |
| RMDS | 56.61 | 67.50 | 73.48 | 74.25 | 72.13 | 62.06 | 73.29 | 68.79 |
| Gram | 51.93 | 60.63 | 71.36 | 84.83 | 69.40 | 56.28 | 75.20 | 67.63 |
| ReAct | 55.34 | 64.51 | 87.93 | 81.08 | 79.34 | 59.93 | 82.78 | 73.64 |
| VIM | 45.88 | 59.12 | 72.22 | 93.09 | 75.01 | 52.50 | 80.10 | 69.06 |
| KNN | 43.78 | 59.86 | 67.79 | 90.29 | 69.98 | 51.82 | 76.02 | 66.34 |
| ASH | 54.66 | 66.38 | 89.23 | 89.53 | 81.47 | 60.52 | 86.75 | 76.25 |
| SHE | **58.15** | 64.27 | 84.71 | 87.48 | 76.92 | 61.21 | 83.04 | 74.31 |
| GEN | 52.95 | 62.73 | 78.47 | 71.82 | 72.62 | 57.84 | 74.31 | 67.72 |
| NAC | 52.48 | 66.49 | 88.92 | 92.77 | 80.76 | 59.48 | 87.48 | 76.28 |
| **KAN** | 55.88 | **69.55** | **91.55** | **93.45** | **82.15** | **62.71** | **89.05** | **78.52** |

Table 3: Tab. Med. Caucasian Eth. as InD (AUROC metric).

| Method | eICU - Eth. |
|---|---|
| MDS | **58.5**$_{\pm2.2}$ |
| RMDS | 51.6$_{\pm1.5}$ |
| KNN | 55.8$_{\pm1.9}$ |
| VIM | 57.3$_{\pm2.3}$ |
| SHE | 50.5$_{\pm1.7}$ |
| KLM | 51.6$_{\pm2.1}$ |
| OpenMax | 48.7$_{\pm0.8}$ |
| **KAN** | **61.4**$_{\pm3.1}$ |

Table 4: Tab. Med. > 70 y.o. as InD (AUROC metric).

| Method | eICU - Age |
|---|---|
| MDS | **50.8**$_{\pm1.1}$ |
| RMDS | 48.3$_{\pm0.7}$ |
| KNN | 49.6$_{\pm0.2}$ |
| VIM | 48.8$_{\pm0.1}$ |
| SHE | **50.4**$_{\pm0.7}$ |
| KLM | **51.0**$_{\pm0.7}$ |
| OpenMax | 48.1$_{\pm0.5}$ |
| **KAN** | 50.5$_{\pm0.5}$ |

Table 5: Tab. Med. Feature multiplication (AUROC metric).

| Method | eICU - Synthetic OOD | | | Avg Overall |
|---|---|---|---|---|
| | $\mathcal{F}=10$ | $\mathcal{F}=100$ | $\mathcal{F}=1000$ | |
| MDS | 59.9$_{\pm1.4}$ | 79.5$_{\pm1.4}$ | 87.5$_{\pm0.9}$ | 75.63$_{\pm1.26}$ |
| RMDS | 51.5$_{\pm1.3}$ | 57.8$_{\pm7.4}$ | 64.0$_{\pm13.0}$ | 57.77$_{\pm8.67}$ |
| KNN | 57.3$_{\pm1.4}$ | 75.4$_{\pm2.2}$ | 86.5$_{\pm1.3}$ | 73.07$_{\pm1.68}$ |
| VIM | 57.9$_{\pm1.6}$ | 77.6$_{\pm1.3}$ | **88.3**$_{\pm0.7}$ | 74.60$_{\pm1.26}$ |
| SHE | 55.7$_{\pm1.3}$ | 71.2$_{\pm2.9}$ | 80.4$_{\pm1.6}$ | 69.10$_{\pm2.05}$ |
| KLM | 54.1$_{\pm0.8}$ | 63.1$_{\pm1.1}$ | 72.1$_{\pm4.2}$ | 63.10$_{\pm2.55}$ |
| OpenMax | 51.0$_{\pm0.7}$ | 56.1$_{\pm2.7}$ | 71.4$_{\pm3.2}$ | 59.50$_{\pm2.45}$ |
| **KAN** | **64.6**$_{\pm2.2}$ | **83.0**$_{\pm2.6}$ | **89.8**$_{\pm1.8}$ | **79.13**$_{\pm2.22}$ |

Table 6: The effect of training dataset sizes on AUROC performance.

| Method | CIFAR-10 benchmark | | | | CIFAR-100 benchmark | | |
|---|---|---|---|---|---|---|---|
| | 100% | 10% | 1% | 0.1% | 100% | 10% | 1% |
| VIM | 91.88$_{\pm0.37}$ | 91.69$_{\pm0.38}$ | 88.67$_{\pm1.29}$ | 76.38$_{\pm3.83}$ | 79.46$_{\pm1.62}$ | 78.83$_{\pm1.67}$ | 67.06$_{\pm2.63}$ |
| KNN | 92.19$_{\pm0.27}$ | 91.72$_{\pm0.28}$ | 88.94$_{\pm0.70}$ | 8.15$_{\pm0.86}$ | 81.66$_{\pm0.87}$ | 80.05$_{\pm0.85}$ | 27.03$_{\pm1.71}$ |
| NAC | 87.05$_{\pm1.14}$ | 89.74$_{\pm0.90}$ | 93.09$_{\pm0.65}$ | 89.29$_{\pm0.78}$ | 80.80$_{\pm0.67}$ | 81.72$_{\pm0.59}$ | 80.97$_{\pm1.09}$ |
| **KAN** | **94.12**$_{\pm0.59}$ | **93.95**$_{\pm0.61}$ | **93.90**$_{\pm0.62}$ | **93.21**$_{\pm0.53}$ | **83.44**$_{\pm1.99}$ | **83.11**$_{\pm2.43}$ | **81.44**$_{\pm1.21}$ |

in higher AUROC values. However, there is an upper limit beyond which further increasing the number of partitions does not lead to performance improvements. The choice of k-means clustering over other methods is justified by its simplicity and excellent scaling performance. Additionally, empirical evidence, as reported in Appendix A.7, demonstrates that the choice of the clustering algorithm does not significantly affect detection performance.

According to the authors of KAN (Liu et al., 2024b), varying the grid size has a similar effect to varying the width and depth of a traditional MLP. A fine-grained grid (higher $G$) should improve the accuracy of the network. In the case of our detector, as reported in Table 7, increasing the density of the grid above a certain threshold does not result in higher OOD detection performance.

Table 7: KAN detector performance *w.r.t* different datasets partitions and grid sizes over CIFAR-10.

| Partitions ($\mathcal{P}$) | AUROC | Grid size ($G$) | AUROC |
|---|---|---|---|
| $\mathcal{P} = 1$ | $46.08_{\pm 15.58}$ | $k = 5$ | $87.20_{\pm 1.52}$ |
| $\mathcal{P} = 5$ | $90.39_{\pm 2.78}$ | $k = 10$ | $91.72_{\pm 0.54}$ |
| $\mathcal{P} = 10$ | $\mathbf{94.12}_{\pm 0.59}$ | $k = 50$ | $93.92_{\pm 0.40}$ |
| $\mathcal{P} = 20$ | $94.10_{\pm 0.62}$ | $k = 100$ | $\mathbf{94.12}_{\pm 0.59}$ |
| $\mathcal{P} = 30$ | $94.05_{\pm 0.53}$ | $k = 200$ | $94.03_{\pm 0.49}$ |

**Splines' smoothing operation.** KANs incorporate splines that perform an essential smoothing operation, which is crucial given the continuous nature of the input space. To demonstrate the superiority of the KAN detector, we implemented a baseline histogram method by replacing all the univariate functions $\phi_{p,q}$ in KAN with simple histograms that record the presence of InD samples in a binary manner. The histogram method achieves an overall AUROC of 85.29%, which is approximately 9% lower than that of the KAN detector, clearly demonstrating the superiority of the KAN's splines approach.

**Partitioning alternatives.** To capture the joint feature distribution, the partitioning method is not the only solution. Another approach is to augment the input features with new features that are combinations of the original ones. This can be efficiently achieved using techniques like Principal Component Analysis (PCA) or autoencoders. PCA provides features that are linear combinations of the original ones, while autoencoders generate features that are non-linear combinations. Although this technique worked well on a toy L-shaped dataset (Appendix A.8), it did not yield the desired results on high-dimensional feature spaces in other benchmarks. It resulted in a lower AUROC compared to the partitioning method.

**Influence of the training task.** Since KANs are differentiable, they can be trained similarly to conventional MLPs using backpropagation. In our approach, the KAN is trained with latent features extracted from the backbone as inputs, and the training task mirrors that of the backbone network, specifically multi-class classification. For more details on the used training parameters see Appendix A.9. Importantly, the training task does not need to directly relate to the OOD detection problem. Similar to the histogram baseline, our primary objective is to *register* all input samples within the correct spline coefficients. Any training task that adjusts the spline coefficients in the vicinity of the InD samples can yield a valid OOD detector.

We experimentally verified our hypothesis by training the KAN using a different loss function and an unrelated task, namely regression to a constant value. The results from this regression task demonstrate that the detector effectively distinguishes between InD and OOD samples. Compared to the KAN trained on the classification task, we observed an improvement of approximately 0.2% in AUROC performance on the image benchmarks. However, on the tabular data benchmarks, the performance decreased by approximately 3%. These findings indicate that while modifying the training task of the detector can still yield satisfactory performance, the extent of this effect appears to be benchmark-dependent.

## 4 RELATED WORK

This section reviews recent advancements in OOD detection, provides an overview of the latest innovations to enhance KAN performance, and explores the diverse sectors where KANs have demonstrated successful applications.

### 4.1 OUT-OF-DISTRIBUTION DETECTION

OOD detection focuses on identifying instances with semantic shifts, a special case of distributional shift. OOD detection methods can be broadly classified into the following categories (Yang et al., 2024). **Classification-based methods** use the output of classification models, such as softmax

scores, to distinguish between InD and OOD samples. Examples include Maximum Softmax Probability (MSP) (Hendrycks & Gimpel, 2017), which uses the softmax score of the predicted class as a confidence score, and ODIN (Liang et al., 2018), which applies temperature scaling and input perturbations to enhance the separability of InD and OOD samples. More recent methods that fall in this category are SCALE (Xu et al., 2024a), ASH (Djurisic et al., 2023), VIM (Wang et al., 2022), and KNN (Sun et al., 2022). Gradient-based methods also belong to this category. Examples include GradNorm (Huang et al., 2021) and NAC (Liu et al., 2024a), which use gradients calculated from the KL divergence between the model's output and a uniform probability distribution. **Density-based methods** model the probability distribution of the training data to identify deviations. This is often achieved using a Gaussian mixture model (Zong et al., 2018) or normalizing flows (Zisselman & Tamar, 2020; Jiang et al., 2022). **Reconstruction-based methods** typically use autoencoders to reconstruct input samples and measure the reconstruction error as a signal for OOD detection (Jiang et al., 2023; Zhou, 2022). **Distance-based methods** rely on distance metrics in the feature space to identify OOD samples. The Mahalanobis distance-based detector (Lee et al., 2018) first models the feature distribution with a class-conditional Gaussian distribution and then it derives the InD score using the Mahalanobis distance between the InD centroids and the input sample. fDBD (Liu & Qin, 2024) measures the distance between the latent feature of the sample and the class decision boundaries. Our method also falls into this category, as it computes the InD score by measuring the distance between the network's regions activated during training (InD regions) and those activated by the test sample.

## 4.2 Kolmogorov-Arnold Networks

The recently introduced KAN (Hou & Zhang, 2024) represents a significant advancement in neural network architectures, offering a potential alternative to traditional MLPs by not only enhancing accuracy but also leading to more interpretable models. As a result, numerous studies have tried to innovate and refine KANs further. For example, many articles replace the spline architecture with more efficient or accurate alternatives such as Chebyshev polynomials (SS et al., 2024), wavelet-based structures (Bozorgasl & Chen, 2024), sinusoidal functions (Reinhardt et al., 2024), and radial basis functions (Li, 2024). Others try to replicate advanced neural network architectures using KAN's characteristics. This includes convolutional neural networks (Bodner et al., 2024) and graph neural networks (Kiamari et al., 2024; Bresson et al., 2024; Zhang & Zhang, 2024), further demonstrating the versatility and potential of KANs. Applications of KANs have rapidly expanded across various domains, including time series analysis (Vaca-Rubio et al., 2024; Xu et al., 2024b), solving ordinary and partial differential equations (Koenig et al., 2024; Wang et al., 2024), hyperspectral image classification (Seydi, 2024; Jamali et al., 2024), and computer vision (Azam & Akhtar, 2024; Li et al., 2024; Cheon, 2024). Additionally, KANs have recently been applied to fields similar to OOD detection, such as abnormality detection (Huang et al., 2024) and AI-generated image detection (Anon & Emon, 2024). These studies leverage the superior accuracy and interpretability of KANs (Liu et al., 2024b) to uncover more complex patterns in the data. While their work focuses on developing robust models that demonstrate KANs' capacity to generalize effectively to unseen samples, they do not address the detection of these samples. In contrast, we present a novel OOD detection method that leverages the unique local plasticity property of KANs, applicable to any backbone architecture.

## 5 Conclusions

This paper introduces a novel approach to OOD detection using KANs, capitalizing on their unique local neuroplasticity property. Our method effectively differentiates between InD and OOD samples by comparing the activation patterns of a trained KAN against its untrained counterpart. The experimental results show that our KAN-based detector reaches SOTA performance across seven benchmarks from two different domains. Importantly, our experiments show that the previous methods suffer from a non-optimal InD dataset size, while our method is unaffected by these perturbations. This makes the KAN detector a robust and versatile method that can maintain high performance across diverse and unpredictable data distributions. Future work will further explore the effect of different training tasks on detection performance.

ACKNOWLEDGMENTS

This work is a result of the joint research project STADT:up (19A22006O). The project is supported by the German Federal Ministry for Economic Affairs and Climate Action (BMWK), based on a decision of the German Bundestag. The authors are solely responsible for the content of this publication.

The authors would like to express their sincere gratitude to Pavel Kolev, Niklas Hanselmann, and Shuxiao Ding for their invaluable assistance in proofreading and providing insightful feedback on this manuscript.

REPRODUCIBILITY STATEMENT

Our implementation adheres rigorously to the benchmark guidelines. Detailed information on the hardware and software utilized is provided in Appendix A.10, while the inference time performance of our method is discussed in Appendix A.11. The settings and hyperparameters for each benchmark are reported in Appendix A.12. Our code is publicly available at the following link: `https://github.com/alessandro-canevaro/KAN-OOD`.

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

# A APPENDIX

## A.1 SCORING AND AGGREGATION FUNCTIONS

The trainable coefficients of the detector networks are initialized randomly. As a result, it may occasionally occur that some of these coefficients are initialized to the exact values they would attain post-training. Consequently, the training procedure does not modify these coefficients, as they are already optimal. This phenomenon can lead to false positives in detection, as both the trained and untrained networks might exhibit the same response to an InD sample. This issue can be mitigated using a scoring function that is more robust to outliers, such as the median. This hypothesis is experimentally validated in Table 8.

The use of the maximum function for aggregation allows us to select the detector closest to the test sample, which intuitively possesses the best information for decision-making. This approach is experimentally verified in Table 9.

Table 8: AUROC performance variation on the CIFAR-10 benchmark for different scoring $F_{\text{score}}$ functions.

| Scoring $F_{\text{score}}$ | AUROC |
|---|---|
| min | $90.84_{\pm 0.35}$ |
| mean | $92.04_{\pm 0.37}$ |
| median | $\mathbf{94.12}_{\pm 0.59}$ |
| max | $54.99_{\pm 12.02}$ |

Table 9: AUROC performance variation on the CIFAR-10 benchmark for different aggregation $F_{\text{agg.}}$ functions.

| Aggregation $F_{\text{agg.}}$ | AUROC |
|---|---|
| min | $91.55_{\pm 0.97}$ |
| mean | $91.99_{\pm 2.30}$ |
| median | $90.31_{\pm 3.28}$ |
| max | $\mathbf{94.12}_{\pm 0.59}$ |

## A.2 DETECTION ON REGRESSION-BASED DATASETS

As the standard benchmarks used in OOD detection mainly focus on classification tasks, we test our method on three additional regression-based datasets: the California Housing dataset (Kelley Pace & Barry, 1997), the Wine Quality dataset (Cortez et al., 2009), and the Friedman synthetic dataset (Friedman, 1991). To generate the InD and OOD partitions and ensure that the OOD samples are semantically different from the InD ones, we thresholded the regression (output) value. The KAN is then directly applied to the raw dataset features, highlighting that the method is not only effective on regression-based tasks but also in the absence of a feature extractor backbone network. This is not possible for other methods such as NAC that require the gradients of the backbone network for detection. Thus, as a baseline detector method, we used KNN, which, according to the results in Section 3.2, is one of the best approaches across all benchmarks. Table 10 reports the detection results in terms of AUROC on the three datasets, showing that the KAN detector outperforms the KNN baseline on all of them.

Table 10: Detection (AUROC) results for regression-based datasets.

| Method | California Housing | Wine Quality | Friedman |
|---|---|---|---|
| KNN | 68.73 | 68.69 | 67.30 |
| **KAN** | **70.53** | **71.32** | **69.42** |

On the California Housing and Wine Quality datasets, we used only one partition $(\mathcal{P})$ for the KAN detector because a value greater than one did not improve performance. This indicates that either the partitioning method does not work well on regression-based datasets, possibly due to poor internal separability of data clusters, or these datasets do not require the detector to capture the joint feature distribution to effectively separate InD and OOD samples. To investigate this observation further, we also tested our method on the Friedman dataset. Here, the regression output is generated by the following non-linear function of the inputs:

$$y(\mathbf{x}) = 10 \cdot \sin(\pi \cdot x_0 \cdot x_1) + 20 \cdot (x_2 - 0.5)^2 + 10 \cdot x_3 + 5 \cdot x_4 + \mathcal{N}(0, \sigma). \tag{8}$$

Given this non-linearity, it is ensured that the samples are not separable using only the marginal feature distribution. In this case, peak performance is achieved with a minimum of four partitions, as shown in Table 11. This shows that even for regression-based datasets our method can capture the joint feature distribution.

Table 11: AUROC performance as a function of the number of partitions ($\mathcal{P}$) in the Friedman dataset.

| Partitions ($\mathcal{P}$) | 1 | 2 | 3 | 4 |
|---|---|---|---|---|
| AUROC | 52.11 | 64.15 | 63.83 | **69.42** |

## A.3 AVERAGE OVERALL METRIC

Many benchmarks (including the OpenOOD CIFAR-10 and CIFAR-100) assess OOD detection performance on multiple OOD datasets. However, they lack an overall average that gives a holistic overview of the methods' performance. In our experiments, we additionally evaluate our method on the following *overall* metric:

$$\mu_{\text{overall}} = \frac{1}{N}\sum_{i=1}^{N}\mu_i, \quad \sigma_{\text{overall}} = \sqrt{\frac{1}{N}\sum_{i=1}^{N}\sigma_i^2} \tag{9}$$

where $\mu_{\text{i}}, \sigma_{\text{i}}$ are the mean and standard deviation of dataset $i$ calculated over multiple seeds.

## A.4 EFFECT OF KAN STOCHASTICITY

All benchmarks average results over multiple seeds to address the inherent randomness associated with weight initialization in the backbone model. Our method introduces an additional layer of randomness due to the KAN initialization process. To illustrate that the variability introduced by our detector is significantly lower than that stemming from the backbone initialization, we conducted the following experiment.

We repeated the CIFAR-10 benchmark using five distinct KAN initialization seeds ($N = 5$). For each KAN initialization seed $i$, we recorded the mean and standard deviation ($\mu_i, \sigma_i$) of the experiment conducted on the three pre-trained backbones specified in the benchmark. The results are summarized in Table 12.

Table 12: CIFAR-10 benchmark results across different KAN initializations.

| KAN seed ($i$) | 1 | 2 | 3 | 4 | 5 |
|---|---|---|---|---|---|
| AUROC ($\mu_i \pm \sigma_i$) | 94.12±0.59 | 94.02±0.58 | 94.11±0.52 | 94.17±0.57 | 94.06±0.39 |

We compute the overall standard deviation attributable to the backbone initialization ($\sigma_{\text{b}}$) and that of our detector ($\sigma_{\text{d}}$) as follows:

$$\sigma_{\text{b}} = \frac{1}{N}\sum_{i=1}^{N}\sigma_i = 0.53, \quad \sigma_{\text{d}} = \sqrt{\frac{1}{N}\sum_{i=1}^{N}(\sigma_i - \mu_{\text{b}})^2} = 0.05 \quad \text{with} \quad \mu_{\text{b}} = \frac{1}{N}\sum_{i=1}^{N}\mu_i = 94.10 \tag{10}$$

The calculated standard deviations $\sigma_{\text{b}}$ and $\sigma_{\text{d}}$ differ by approximately an order of magnitude, indicating that the randomness introduced by our detector has a negligible effect on the overall results.

## A.5 BASELINES METHODS

The baselines used in the banchmarks are: OpenMax (Bendale & Boult, 2016), MSP (Hendrycks & Gimpel, 2017), TempScale (Guo et al., 2017), ODIN (Liang et al., 2018), MDS (Lee et al., 2018),

MDSEns (Lee et al., 2018), RMDS (Ren et al., 2021), Gram (Sastry & Oore, 2020), EBO (Liu et al., 2020), OpenGAN (Kong & Ramanan, 2021), GradNorm (Huang et al., 2021), ReAct (Sun et al., 2021), MLS (Hendrycks et al., 2022), KLM (Hendrycks et al., 2022), VIM (Wang et al., 2022), KNN (Sun et al., 2022), DICE (Sun & Li, 2022), RankFeat (Song et al., 2022), ASH (Djurisic et al., 2023), SHE (Zhang et al., 2023b), GEN (Liu et al., 2023), and NAC (Liu et al., 2024a).

## A.6 FULL BENCHMARK RESULTS

In Table 17, we present the AUROC results for all officially available baselines on the CIFAR-10 and CIFAR-100 benchmarks. Table 22 provides the results for the same set of baselines and benchmarks using the FPR@95 metric. Similarly, Tables 18 and 23 report the AUROC and FPR@95 results respectively for all the available baselines on the ImageNet-200 FS and ImageNet-1K FS benchmarks. The same metrics are reported for all the available baselines for the tabular medical benchmarks in Tables 19, 20, 21 for AUROC and 24, 25, 26 for FPR@95.

## A.7 INFLUENCE OF CLUSTERING METHOD

This experiment analyzes the effect of different clustering methods on detection performance. We considered five popular clustering approaches as alternatives to k-means: spectral (Ng et al., 2001), agglomerative (Murtagh & Legendre, 2014), bisecting k-means (Rohilla et al., 2019), BIRCH (Zhang et al., 1996), and DBSCAN (Ester et al., 1996). Table 13 presents the experimental results for the CIFAR-10 benchmark.

Table 13: Detection performance with different clustering algorithms on the CIFAR-10 benchmark.

|  | k-means | spectral | agglomerative | bisecting k-means | BIRCH | DBSCAN |
|---|---|---|---|---|---|---|
| **AUROC** | $94.12_{\pm 0.59}$ | $94.11_{\pm 0.59}$ | $94.12_{\pm 0.59}$ | $94.10_{\pm 0.57}$ | $94.12_{\pm 0.59}$ | $89.82_{\pm 4.64}$ |

The results show that the choice of clustering method has a negligible impact on detection performance, except for DBSCAN, which yields an approximate $4\%$ drop. One reason for this behavior is that DBSCAN is the only algorithm among those considered that does not necessarily assign a cluster to all samples. In our implementation, these unclustered samples are grouped into an additional cluster. However, the samples in this extra cluster do not share common characteristics and can belong to different and distant regions of the input space. Although we are not focused on obtaining semantically meaningful clusters, we aim to divide the InD samples into smaller regions that can be effectively processed by a KAN. The leftover samples cluster in DBSCAN has a counterproductive effect, as it can span a wide region of the input space, making it difficult for the KAN to handle effectively.

## A.8 CAPTURING THE JOINT FEATURE DISTRIBUTION

An alternative approach to the partitioning method for capturing the joint feature distribution is to expand the input features with additional values. We applied this technique to the 2D L-shaped toy dataset, as illustrated in Figure 5. In this scenario, the two input features are concatenated with the latent features derived from a variational autoencoder trained on the two original features, resulting in an augmented input space of size $2 + 64$.

This method demonstrates promising results, comparable to those achieved with the partitioning method. However, its applicability to high-dimensional input spaces remains uncertain. We hypothesize that the number of required features would become excessively large, leading to computational inefficiencies.

## A.9 TRAINING PARAMETERS

Finding the optimal training hyperparameters for KANs can initially be challenging, as they may not follow the same intuitions as MLPs and other networks (Liu et al., 2024b).

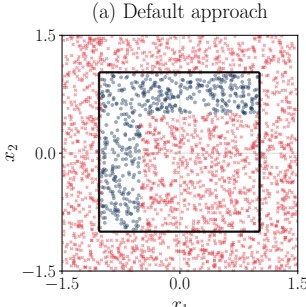
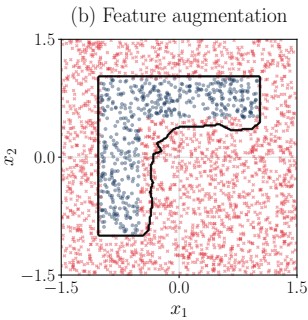

(a) Default approach    (b) Feature augmentation

Figure 5: 2D L-shape toy dataset: The blue point cloud shows the training distribution and the red points are OOD test samples. The black contour represents the thresholded score function (median) separating InD from OOD samples. (a) Default KAN detector limited by the marginal distribution of $x_1$ and $x_2$. (b) Improved performance by concatenating the input features with the latent features of a variational autoencoder.

In our experiments, we use a learning rate of $0.1$ and limit the training to a single epoch. With these settings, the trained KAN achieved a classification accuracy on the CIFAR-10 dataset of approximately $94.50\%$, comparable to the one of the ResNet-18 of $95.06\%$.

To enhance memory efficiency, we employed the AdamW optimizer (Loshchilov & Hutter, 2019) instead of the LBFGS optimizer (Liu & Nocedal, 1989) originally suggested by the KAN authors.

### A.10 SOFTWARE AND HARDWARE

All experiments are performed on a single NVIDIA GeForce GTX 1080Ti GPU. For testing larger models and accelerating the hyperparameter optimization, we leveraged a cloud computing platform with an NVIDIA A100 GPU.

We used Python version 3.10 together with PyTorch 2.3.1 as the deep learning framework and leveraged CUDA 11.8 for GPU acceleration.

### A.11 INFERENCE TIME AND SCALABILITY ANALYSIS

**Inference time.** Table 14 reports the inference time of a single sample for various methods. The measurements are averaged over 1000 samples, using 100 extra samples as GPU warmup. The results show a positive correlation between the inference time and the overall AUROC performance.

Although the KAN method is currently the slowest among the tested ones, it is important to emphasize that the KAN architecture has just been developed recently. In just a few months since its release, its performance has been steadily improving thanks to many architecture refinements (e.g., replacing splines with Gaussian radial basis functions improves forward speed by approximately a factor of 3.3 (Li, 2024)). We believe that in the future, KANs will achieve efficiency comparable to MLPs. Furthermore, it is worth considering that inference time is not always a critical concern in various applications, particularly in medical contexts. In such scenarios, the enhanced detection performance offered by our method positions it as a more advantageous choice compared to faster alternatives.

Table 14: Inference time of single sample compared to the overall AUROC on CIFAR10.

| Method | Inference time (ms) | Overall AUROC |
|--------|--------------------|--------------:|
| VIM    | 0.271              | 91.88         |
| KNN    | 0.175              | 92.19         |
| NAC    | 0.681              | 93.37         |
| KAN    | 2.216              | 94.12         |

**Setup time.** In Table 15, we report the setup time of our detector for different training dataset sizes and various numbers of partitions ($\mathcal{P}$), using the KNN method as a reference baseline.

Table 15: Setup time in seconds for different training dataset sizes and number of partitions.

| Method | 10K | 100K | 1M |
|---|---|---|---|
| KAN - $\mathcal{P} = 1$ | 2.84 | 17.49 | 165.86 |
| KAN - $\mathcal{P} = 10$ | 2.82 | 17.44 | 166.45 |
| KAN - $\mathcal{P} = 100$ | 3.19 | 18.24 | 172.66 |
| KNN | 3.27 | 15.51 | 141.75 |

The results indicate that the biggest factor influencing setup time (which includes inference on the backbone model, the partitioning method and the training of the KANs) is the dataset size; however, our method shows comparable speeds to the KNN baseline. On the other hand, varying the number of partitions seems to have a smaller influence, likely due to GPU parallelization.

**Further considerations.** The number of parameters in the KAN network is determined by the product of three factors: the number of inputs, the number of outputs, and the grid size. The grid size depends on the specific benchmark, and our experiments indicate that it does not correlate with the benchmark's complexity. For instance, on CIFAR-10 the optimal value is 100, while on CIFAR-100 it is 50. Scalability to larger images, or more generally to large input spaces, is typically not an issue as our detector is applied to the latent space of the backbone model, which is usually much smaller than the inputs. For example, ImageNet-200 has an input space of roughly 150k dimensions, but it is compressed by the backbone into a latent space of just 512 features. Lastly, our preliminary results presented in Section 3.3 demonstrate that changing the training task of the KAN detector can lead to similar performance. This can can be used to reduce the number of outputs required by the KAN model, further improving scalability. For example, in the ImageNet-1K FS benchmark, we employed class-based partitioning, resulting in 1000 clusters and, consequently, 1000 models. However, we reduced the number of outputs for each model from 1000 to 10 classes by randomly grouping labels together. This adjustment is motivated by the fact that with 1000 clusters, the problem tackled by each model is greatly reduced, and thus the model's capacity can also be reduced. As a result, the training time per sample of our model is slightly lower than that for the ImageNet-200 FS benchmark: approximately 1.6ms per sample compared to 1.9ms per sample for ImageNet-200 FS. This indicates that our method remains efficient and robust even with a large number of clusters.

## A.12 HYPERPARAMETERS

Table 16 reports all hyperparameters and settings for the five benchmarks. The search space of each hyperparameter is as follows: [10, 200] for the grid size, [1, 200] for the partitions, [0.0001, 0.1] for the learning rate, [1, 100] for the epochs, and [1, 100] for the histogram bins.

Table 16: Hyperparameters.

| Parameter | OpenOOD | | | | TabMed | | |
|---|---|---|---|---|---|---|---|
| | CIFAR10 | CIFAR100 | ImageNet-200 FS | ImageNet-1K FS | Ethnicity | Age | Synthetic |
| Grid size | 100 | 50 | 10 | 50 | 100 | 50 | 50 |
| Partitions | 10 | 100 | 200 | 1000 | 30 | 30 | 10 |
| Train Size | 100% | 100% | 100% | 100% | 100% | 100% | 100% |
| Learning rate | 0.1 | 0.1 | 0.1 | 0.1 | 0.1 | 0.1 | 0.1 |
| Epochs | 1 | 1 | 1 | 1 | 1 | 1 | 1 |
| Partitioning method | k-means | class-based | class-based | class-based | k-means | k-means | k-means |
| Histogram norm. | yes | yes | yes | yes | yes | no | no |
| Histogram bins | 5 | 10 | 5 | 5 | 40 | - | - |
| Backbone | ResNet-18 | ResNet-18 | ResNet-18 | ResNet-50 | FT-Transformer | FT-Transformer | FT-Transformer |
| Pre-Trained backbone | yes | yes | yes | yes | no | no | no |

Table 17: AUROC performance on CIFAR-10 and CIFAR-100 benchmarks.

| Method | Near OOD | | Far OOD | | | | Avg Near | Avg Far | Avg Overall |
|---|---|---|---|---|---|---|---|---|---|
| | CIFAR | TIN | MNIST | SVHN | Textures | Places365 | | | |
| **CIFAR-10 Benchmark** | | | | | | | | | |
| OpenMax | $86.91_{\pm0.31}$ | $88.32_{\pm0.28}$ | $90.50_{\pm0.44}$ | $89.77_{\pm0.45}$ | $89.58_{\pm0.60}$ | $88.63_{\pm0.28}$ | $87.62_{\pm0.29}$ | $89.62_{\pm0.19}$ | $88.95_{\pm0.41}$ |
| MSP | $87.19_{\pm0.33}$ | $88.87_{\pm0.19}$ | $92.63_{\pm1.57}$ | $91.46_{\pm0.40}$ | $89.89_{\pm0.71}$ | $88.92_{\pm0.47}$ | $88.03_{\pm0.25}$ | $90.73_{\pm0.43}$ | $89.83_{\pm0.76}$ |
| TempScale | $87.17_{\pm0.40}$ | $89.00_{\pm0.23}$ | $93.11_{\pm1.77}$ | $91.66_{\pm0.52}$ | $90.01_{\pm0.74}$ | $89.11_{\pm0.52}$ | $88.09_{\pm0.31}$ | $90.97_{\pm0.52}$ | $90.01_{\pm0.86}$ |
| ODIN | $82.18_{\pm1.87}$ | $83.55_{\pm1.84}$ | $\mathbf{95.24}_{\pm1.96}$ | $84.58_{\pm0.77}$ | $86.94_{\pm2.26}$ | $85.07_{\pm1.24}$ | $82.87_{\pm1.85}$ | $87.96_{\pm0.61}$ | $86.26_{\pm1.73}$ |
| MDS | $83.59_{\pm2.27}$ | $84.81_{\pm2.53}$ | $90.10_{\pm2.41}$ | $91.18_{\pm0.47}$ | $92.69_{\pm1.06}$ | $84.90_{\pm2.54}$ | $84.20_{\pm2.40}$ | $89.72_{\pm1.36}$ | $87.88_{\pm2.05}$ |
| MDSEns | $61.29_{\pm0.23}$ | $59.57_{\pm0.53}$ | $\mathbf{99.17}_{\pm0.41}$ | $66.56_{\pm0.58}$ | $77.40_{\pm0.28}$ | $52.47_{\pm0.15}$ | $60.43_{\pm0.26}$ | $73.90_{\pm0.27}$ | $69.41_{\pm0.40}$ |
| RMDS | $88.83_{\pm0.35}$ | $90.76_{\pm0.27}$ | $93.22_{\pm0.80}$ | $91.84_{\pm0.26}$ | $92.23_{\pm0.23}$ | $\mathbf{91.51}_{\pm0.11}$ | $89.80_{\pm0.28}$ | $92.20_{\pm0.21}$ | $91.40_{\pm0.40}$ |
| Gram | $58.33_{\pm4.49}$ | $58.98_{\pm5.19}$ | $72.64_{\pm2.34}$ | $\mathbf{91.52}_{\pm4.45}$ | $62.34_{\pm8.27}$ | $60.44_{\pm3.41}$ | $58.66_{\pm4.83}$ | $71.73_{\pm3.20}$ | $67.37_{\pm5.04}$ |
| EBO | $86.36_{\pm0.58}$ | $88.80_{\pm0.36}$ | $\mathbf{94.32}_{\pm2.53}$ | $91.79_{\pm0.98}$ | $89.47_{\pm0.70}$ | $89.25_{\pm0.78}$ | $87.58_{\pm0.46}$ | $91.21_{\pm0.92}$ | $90.00_{\pm1.22}$ |
| OpenGAN | $52.81_{\pm7.69}$ | $54.62_{\pm7.68}$ | $\mathbf{56.14}_{\pm24.08}$ | $\mathbf{52.81}_{\pm27.60}$ | $\mathbf{56.14}_{\pm18.26}$ | $53.34_{\pm5.79}$ | $53.71_{\pm7.68}$ | $54.61_{\pm15.51}$ | $\mathbf{54.31}_{\pm17.45}$ |
| GradNorm | $54.43_{\pm1.59}$ | $55.37_{\pm0.41}$ | $63.72_{\pm7.37}$ | $53.91_{\pm6.36}$ | $52.07_{\pm4.09}$ | $60.50_{\pm5.33}$ | $54.90_{\pm0.98}$ | $57.55_{\pm3.22}$ | $56.67_{\pm4.88}$ |
| ReAct | $85.93_{\pm0.83}$ | $88.29_{\pm0.44}$ | $\mathbf{92.81}_{\pm3.03}$ | $89.12_{\pm3.19}$ | $89.38_{\pm1.49}$ | $\mathbf{90.35}_{\pm0.78}$ | $87.11_{\pm0.61}$ | $90.42_{\pm1.41}$ | $89.31_{\pm1.96}$ |
| MLS | $86.31_{\pm0.59}$ | $88.72_{\pm0.36}$ | $\mathbf{94.15}_{\pm2.48}$ | $91.69_{\pm0.94}$ | $89.41_{\pm0.71}$ | $89.14_{\pm0.76}$ | $87.52_{\pm0.47}$ | $91.10_{\pm0.89}$ | $89.90_{\pm1.20}$ |
| KLM | $77.89_{\pm0.75}$ | $80.49_{\pm0.85}$ | $85.00_{\pm2.04}$ | $84.99_{\pm1.18}$ | $82.35_{\pm0.33}$ | $78.37_{\pm0.33}$ | $79.19_{\pm0.80}$ | $82.68_{\pm0.21}$ | $81.52_{\pm1.08}$ |
| VIM | $87.75_{\pm0.28}$ | $89.62_{\pm0.33}$ | $94.76_{\pm0.38}$ | $94.50_{\pm0.48}$ | $\mathbf{95.15}_{\pm0.34}$ | $89.49_{\pm0.39}$ | $88.68_{\pm0.28}$ | $93.48_{\pm0.24}$ | $91.88_{\pm0.37}$ |
| KNN | $\mathbf{89.73}_{\pm0.14}$ | $91.56_{\pm0.26}$ | $94.26_{\pm0.38}$ | $92.67_{\pm0.30}$ | $93.16_{\pm0.24}$ | $91.77_{\pm0.23}$ | $90.64_{\pm0.20}$ | $92.96_{\pm0.14}$ | $92.19_{\pm0.27}$ |
| DICE | $77.01_{\pm0.88}$ | $79.67_{\pm0.87}$ | $\mathbf{90.37}_{\pm5.97}$ | $90.02_{\pm1.77}$ | $81.86_{\pm2.35}$ | $74.67_{\pm4.98}$ | $78.34_{\pm0.79}$ | $84.23_{\pm1.89}$ | $82.27_{\pm3.43}$ |
| RankFeat | $77.98_{\pm2.24}$ | $80.94_{\pm2.80}$ | $75.87_{\pm5.22}$ | $68.15_{\pm7.44}$ | $73.46_{\pm6.49}$ | $\mathbf{85.99}_{\pm3.04}$ | $79.46_{\pm2.52}$ | $75.87_{\pm5.06}$ | $77.07_{\pm4.95}$ |
| ASH | $74.11_{\pm1.55}$ | $76.44_{\pm0.61}$ | $83.16_{\pm4.66}$ | $73.46_{\pm6.41}$ | $77.45_{\pm2.39}$ | $79.89_{\pm3.69}$ | $75.27_{\pm1.04}$ | $78.49_{\pm2.58}$ | $77.42_{\pm3.76}$ |
| SHE | $80.31_{\pm0.69}$ | $82.76_{\pm0.43}$ | $\mathbf{90.43}_{\pm4.76}$ | $86.38_{\pm1.32}$ | $81.57_{\pm1.21}$ | $82.89_{\pm1.22}$ | $81.54_{\pm0.51}$ | $85.32_{\pm1.43}$ | $84.06_{\pm2.16}$ |
| GEN | $87.21_{\pm0.36}$ | $89.20_{\pm0.25}$ | $93.83_{\pm2.14}$ | $91.97_{\pm0.66}$ | $90.14_{\pm0.76}$ | $89.46_{\pm0.65}$ | $88.20_{\pm0.30}$ | $91.35_{\pm0.69}$ | $90.30_{\pm1.02}$ |
| NAC | $\mathbf{89.83}_{\pm0.29}$ | $\mathbf{92.02}_{\pm0.20}$ | $94.86_{\pm1.37}$ | $96.06_{\pm0.47}$ | $95.64_{\pm0.45}$ | $91.85_{\pm0.28}$ | $90.93_{\pm0.23}$ | $94.60_{\pm0.50}$ | $93.37_{\pm0.64}$ |
| **KAN** | $\mathbf{90.06}_{\pm0.47}$ | $91.92_{\pm0.52}$ | $\mathbf{97.86}_{\pm0.73}$ | $\mathbf{97.39}_{\pm0.42}$ | $\mathbf{95.85}_{\pm0.28}$ | $91.64_{\pm0.91}$ | $\mathbf{90.99}_{\pm0.50}$ | $\mathbf{95.69}_{\pm0.22}$ | $\mathbf{94.12}_{\pm0.40}$ |
| **CIFAR-100 Benchmark** | | | | | | | | | |
| OpenMax | $74.38_{\pm0.37}$ | $78.44_{\pm0.14}$ | $76.01_{\pm1.39}$ | $82.07_{\pm1.53}$ | $80.56_{\pm0.09}$ | $79.29_{\pm0.40}$ | $76.41_{\pm0.25}$ | $79.48_{\pm0.41}$ | $78.46_{\pm0.88}$ |
| MSP | $78.47_{\pm0.07}$ | $82.07_{\pm0.17}$ | $76.08_{\pm1.86}$ | $78.42_{\pm0.89}$ | $77.32_{\pm0.71}$ | $79.22_{\pm0.29}$ | $80.27_{\pm0.11}$ | $77.76_{\pm0.44}$ | $78.60_{\pm0.90}$ |
| TempScale | $79.02_{\pm0.06}$ | $82.79_{\pm0.09}$ | $77.27_{\pm1.85}$ | $79.79_{\pm1.05}$ | $78.11_{\pm0.72}$ | $79.80_{\pm0.25}$ | $80.90_{\pm0.07}$ | $78.74_{\pm0.51}$ | $\mathbf{79.46}_{\pm0.92}$ |
| ODIN | $78.18_{\pm0.14}$ | $81.63_{\pm0.08}$ | $83.79_{\pm1.31}$ | $74.54_{\pm0.76}$ | $79.33_{\pm1.08}$ | $79.45_{\pm0.26}$ | $79.90_{\pm0.11}$ | $79.28_{\pm0.21}$ | $\mathbf{79.49}_{\pm0.77}$ |
| MDS | $55.87_{\pm0.22}$ | $61.50_{\pm0.28}$ | $67.47_{\pm0.81}$ | $70.68_{\pm6.40}$ | $76.26_{\pm0.69}$ | $63.15_{\pm0.49}$ | $58.69_{\pm0.09}$ | $69.39_{\pm1.39}$ | $65.82_{\pm2.66}$ |
| MDSEns | $43.85_{\pm0.31}$ | $48.78_{\pm0.19}$ | $\mathbf{98.21}_{\pm0.78}$ | $53.76_{\pm1.63}$ | $69.75_{\pm1.14}$ | $42.27_{\pm0.73}$ | $46.31_{\pm0.24}$ | $66.00_{\pm0.69}$ | $59.44_{\pm0.93}$ |
| RMDS | $77.75_{\pm0.19}$ | $82.55_{\pm0.02}$ | $79.74_{\pm2.49}$ | $84.89_{\pm1.10}$ | $83.65_{\pm0.51}$ | $\mathbf{83.40}_{\pm0.46}$ | $80.15_{\pm0.11}$ | $82.92_{\pm0.42}$ | $\mathbf{82.00}_{\pm1.15}$ |
| Gram | $49.41_{\pm0.94}$ | $53.91_{\pm1.58}$ | $80.71_{\pm4.15}$ | $\mathbf{95.55}_{\pm0.60}$ | $70.79_{\pm1.32}$ | $46.38_{\pm1.32}$ | $51.66_{\pm0.77}$ | $73.36_{\pm1.08}$ | $66.12_{\pm1.98}$ |
| EBO | $79.05_{\pm0.11}$ | $82.76_{\pm0.08}$ | $79.18_{\pm1.37}$ | $82.03_{\pm1.74}$ | $78.35_{\pm0.83}$ | $79.52_{\pm0.23}$ | $80.91_{\pm0.08}$ | $79.77_{\pm0.61}$ | $\mathbf{80.15}_{\pm0.97}$ |
| OpenGAN | $63.23_{\pm2.44}$ | $68.74_{\pm2.29}$ | $\mathbf{68.14}_{\pm18.78}$ | $68.40_{\pm2.15}$ | $65.84_{\pm3.43}$ | $69.13_{\pm7.08}$ | $65.98_{\pm1.26}$ | $67.88_{\pm7.16}$ | $67.25_{\pm8.47}$ |
| GradNorm | $70.32_{\pm0.20}$ | $69.95_{\pm0.79}$ | $65.35_{\pm1.12}$ | $76.95_{\pm4.73}$ | $64.58_{\pm0.13}$ | $69.69_{\pm0.17}$ | $70.13_{\pm0.47}$ | $69.14_{\pm1.05}$ | $69.47_{\pm2.01}$ |
| ReAct | $78.65_{\pm0.05}$ | $82.88_{\pm0.08}$ | $78.37_{\pm1.59}$ | $83.01_{\pm0.97}$ | $80.15_{\pm0.46}$ | $80.03_{\pm0.11}$ | $80.77_{\pm0.05}$ | $80.39_{\pm0.49}$ | $\mathbf{80.52}_{\pm0.79}$ |
| MLS | $\mathbf{79.21}_{\pm0.10}$ | $82.90_{\pm0.05}$ | $78.91_{\pm1.47}$ | $81.65_{\pm1.49}$ | $78.39_{\pm0.84}$ | $79.75_{\pm0.24}$ | $81.05_{\pm0.07}$ | $79.67_{\pm0.57}$ | $80.14_{\pm0.93}$ |
| KLM | $73.91_{\pm0.25}$ | $79.22_{\pm0.28}$ | $74.15_{\pm2.59}$ | $79.34_{\pm0.44}$ | $75.77_{\pm0.45}$ | $75.70_{\pm0.24}$ | $76.56_{\pm0.25}$ | $76.24_{\pm0.52}$ | $76.35_{\pm1.10}$ |
| VIM | $72.21_{\pm0.41}$ | $77.76_{\pm0.16}$ | $81.89_{\pm1.02}$ | $83.14_{\pm3.71}$ | $\mathbf{85.91}_{\pm0.78}$ | $75.85_{\pm0.37}$ | $74.98_{\pm0.13}$ | $81.70_{\pm0.62}$ | $\mathbf{79.46}_{\pm1.62}$ |
| KNN | $77.02_{\pm0.25}$ | $\mathbf{83.34}_{\pm0.16}$ | $82.36_{\pm1.52}$ | $84.15_{\pm1.09}$ | $83.66_{\pm0.83}$ | $79.43_{\pm0.47}$ | $80.18_{\pm0.15}$ | $82.40_{\pm0.17}$ | $\mathbf{81.66}_{\pm0.87}$ |
| DICE | $78.04_{\pm0.32}$ | $80.72_{\pm0.30}$ | $79.86_{\pm1.89}$ | $84.22_{\pm2.00}$ | $77.63_{\pm0.34}$ | $78.33_{\pm0.66}$ | $79.38_{\pm0.23}$ | $80.01_{\pm0.18}$ | $\mathbf{79.80}_{\pm1.18}$ |
| RankFeat | $58.04_{\pm2.36}$ | $65.72_{\pm0.22}$ | $63.03_{\pm3.86}$ | $72.14_{\pm1.39}$ | $69.40_{\pm3.08}$ | $63.82_{\pm1.83}$ | $61.88_{\pm1.28}$ | $67.10_{\pm1.42}$ | $65.36_{\pm2.43}$ |
| ASH | $76.48_{\pm0.30}$ | $79.92_{\pm0.20}$ | $77.23_{\pm0.46}$ | $85.60_{\pm1.40}$ | $80.72_{\pm0.70}$ | $80.16_{\pm0.15}$ | $78.20_{\pm0.15}$ | $80.58_{\pm0.66}$ | $\mathbf{79.79}_{\pm0.69}$ |
| SHE | $78.15_{\pm0.03}$ | $79.74_{\pm0.36}$ | $76.76_{\pm1.07}$ | $80.97_{\pm3.98}$ | $73.64_{\pm1.28}$ | $76.30_{\pm0.51}$ | $78.95_{\pm0.18}$ | $76.92_{\pm1.16}$ | $77.59_{\pm1.78}$ |
| GEN | $\mathbf{79.38}_{\pm0.04}$ | $\mathbf{83.25}_{\pm0.13}$ | $78.29_{\pm2.05}$ | $81.41_{\pm1.50}$ | $78.74_{\pm0.81}$ | $80.28_{\pm0.27}$ | $\mathbf{81.31}_{\pm0.08}$ | $79.68_{\pm0.75}$ | $\mathbf{80.23}_{\pm1.10}$ |
| NAC | $72.02_{\pm0.69}$ | $79.86_{\pm0.23}$ | $93.26_{\pm1.34}$ | $92.60_{\pm1.14}$ | $\mathbf{89.36}_{\pm0.54}$ | $73.06_{\pm0.63}$ | $75.94_{\pm0.41}$ | $\mathbf{87.07}_{\pm0.30}$ | $83.36_{\pm0.84}$ |
| **KAN** | $72.97_{\pm0.17}$ | $81.37_{\pm0.22}$ | $92.29_{\pm1.85}$ | $\mathbf{87.16}_{\pm4.46}$ | $\mathbf{89.43}_{\pm0.39}$ | $77.42_{\pm0.35}$ | $77.17_{\pm0.17}$ | $\mathbf{86.57}_{\pm0.70}$ | $\mathbf{83.44}_{\pm1.99}$ |

Table 18: AUROC performance on ImageNet-200 FS and ImageNet-1K FS benchmarks.

| Method | Near OOD | | Far OOD | | | Avg Near | Avg Far | Avg Overall |
|---|---|---|---|---|---|---|---|---|
| | SSB-hard | NINCO | iNaturalist | Textures | OpenImage-O | | | |
| **ImageNet-200 FS Benchmark** | | | | | | | | |
| OpenMax | 47.64$_{\pm0.20}$ | 54.15$_{\pm0.23}$ | 72.44$_{\pm0.87}$ | 69.12$_{\pm0.36}$ | 62.31$_{\pm0.24}$ | 50.89$_{\pm0.18}$ | 67.96$_{\pm0.39}$ | 61.13$_{\pm0.46}$ |
| MSP | 50.94$_{\pm0.25}$ | 57.76$_{\pm0.46}$ | 70.42$_{\pm0.67}$ | 65.11$_{\pm0.43}$ | 62.80$_{\pm0.24}$ | 54.35$_{\pm0.35}$ | 66.11$_{\pm0.37}$ | 61.41$_{\pm0.44}$ |
| TempScale | 50.05$_{\pm0.28}$ | 56.86$_{\pm0.47}$ | 70.18$_{\pm0.76}$ | 65.65$_{\pm0.43}$ | 62.29$_{\pm0.26}$ | 53.46$_{\pm0.37}$ | 66.04$_{\pm0.41}$ | 61.01$_{\pm0.48}$ |
| ODIN | 44.31$_{\pm0.02}$ | 52.36$_{\pm0.08}$ | 70.19$_{\pm0.92}$ | 67.10$_{\pm0.34}$ | 61.48$_{\pm0.31}$ | 48.33$_{\pm0.05}$ | 66.25$_{\pm0.42}$ | 59.09$_{\pm0.46}$ |
| MDS | 48.59$_{\pm0.88}$ | 56.65$_{\pm0.94}$ | 68.25$_{\pm1.51}$ | 73.84$_{\pm0.75}$ | 61.90$_{\pm0.57}$ | 52.62$_{\pm0.90}$ | 68.00$_{\pm0.87}$ | 61.85$_{\pm0.98}$ |
| MDSEns | 34.22$_{\pm0.44}$ | 41.58$_{\pm0.17}$ | 43.63$_{\pm0.48}$ | 67.54$_{\pm0.35}$ | 48.38$_{\pm0.36}$ | 37.90$_{\pm0.20}$ | 53.18$_{\pm0.39}$ | 47.07$_{\pm0.38}$ |
| RMDS | 56.24$_{\pm0.62}$ | 60.95$_{\pm0.94}$ | 71.71$_{\pm1.49}$ | 64.61$_{\pm1.07}$ | 63.52$_{\pm0.83}$ | 58.59$_{\pm0.77}$ | 66.62$_{\pm1.11}$ | 63.41$_{\pm1.03}$ |
| Gram | **59.12**$_{\pm0.73}$ | **63.35**$_{\pm0.76}$ | 58.42$_{\pm0.75}$ | 75.86$_{\pm0.10}$ | 61.51$_{\pm0.39}$ | **61.23**$_{\pm0.74}$ | 65.26$_{\pm0.31}$ | 63.65$_{\pm0.61}$ |
| EBO | 47.56$_{\pm0.29}$ | 53.45$_{\pm0.32}$ | 65.91$_{\pm1.41}$ | 68.03$_{\pm0.48}$ | 59.83$_{\pm0.59}$ | 50.51$_{\pm0.30}$ | 64.59$_{\pm0.76}$ | 58.96$_{\pm0.74}$ |
| OpenGAN | 41.58$_{\pm5.51}$ | 46.51$_{\pm2.97}$ | 61.78$_{\pm8.75}$ | 59.02$_{\pm4.68}$ | 58.53$_{\pm3.57}$ | 44.04$_{\pm4.21}$ | 59.78$_{\pm4.65}$ | 53.48$_{\pm5.48}$ |
| GradNorm | 53.50$_{\pm1.03}$ | 55.32$_{\pm0.88}$ | 70.38$_{\pm2.68}$ | 73.12$_{\pm0.59}$ | 63.52$_{\pm1.33}$ | 54.41$_{\pm0.94}$ | 69.01$_{\pm1.39}$ | 63.17$_{\pm1.49}$ |
| ReAct | 47.25$_{\pm0.57}$ | 53.84$_{\pm0.55}$ | 69.45$_{\pm3.94}$ | 71.45$_{\pm2.04}$ | 62.30$_{\pm2.32}$ | 50.55$_{\pm0.19}$ | 67.73$_{\pm2.76}$ | 60.86$_{\pm2.27}$ |
| MLS | 48.18$_{\pm0.28}$ | 54.43$_{\pm0.32}$ | 67.94$_{\pm1.20}$ | 67.12$_{\pm0.45}$ | 60.66$_{\pm0.44}$ | 51.30$_{\pm0.30}$ | 65.24$_{\pm0.61}$ | 59.67$_{\pm0.64}$ |
| KLM | 54.03$_{\pm0.71}$ | 60.52$_{\pm0.13}$ | 75.37$_{\pm0.91}$ | 64.12$_{\pm0.65}$ | 66.20$_{\pm0.29}$ | 57.28$_{\pm0.41}$ | 68.56$_{\pm0.52}$ | 64.05$_{\pm0.61}$ |
| VIM | 45.34$_{\pm0.72}$ | 57.09$_{\pm1.03}$ | 71.34$_{\pm1.68}$ | **82.54**$_{\pm0.73}$ | 65.70$_{\pm0.94}$ | 51.22$_{\pm0.86}$ | 73.19$_{\pm1.10}$ | 64.40$_{\pm1.08}$ |
| KNN | 44.05$_{\pm0.42}$ | 54.51$_{\pm0.62}$ | 71.53$_{\pm1.32}$ | 81.88$_{\pm0.19}$ | 62.12$_{\pm0.79}$ | 49.28$_{\pm0.51}$ | 71.84$_{\pm0.72}$ | 62.82$_{\pm0.77}$ |
| DICE | 48.11$_{\pm0.51}$ | 54.31$_{\pm0.55}$ | 66.53$_{\pm2.14}$ | 72.51$_{\pm0.66}$ | 61.73$_{\pm0.94}$ | 51.21$_{\pm0.53}$ | 66.92$_{\pm1.12}$ | 60.64$_{\pm1.14}$ |
| RankFeat | 46.41$_{\pm0.23}$ | 43.03$_{\pm2.29}$ | 22.74$_{\pm4.50}$ | 20.60$_{\pm3.08}$ | 40.74$_{\pm4.41}$ | 44.72$_{\pm1.07}$ | 28.03$_{\pm3.96}$ | 34.70$_{\pm3.30}$ |
| ASH | 50.96$_{\pm0.93}$ | 58.51$_{\pm0.60}$ | 77.96$_{\pm1.58}$ | 79.39$_{\pm0.61}$ | **69.09**$_{\pm0.71}$ | 54.74$_{\pm0.74}$ | 75.48$_{\pm0.95}$ | 67.18$_{\pm0.96}$ |
| SHE | 52.82$_{\pm0.65}$ | 56.64$_{\pm0.69}$ | 72.20$_{\pm2.65}$ | 74.27$_{\pm0.63}$ | 64.95$_{\pm1.25}$ | 54.73$_{\pm0.67}$ | 70.47$_{\pm1.39}$ | 64.18$_{\pm1.41}$ |
| GEN | 48.33$_{\pm0.27}$ | 54.85$_{\pm0.42}$ | 68.94$_{\pm0.63}$ | 66.58$_{\pm0.47}$ | 60.87$_{\pm0.28}$ | 51.59$_{\pm0.34}$ | 65.46$_{\pm0.44}$ | 59.91$_{\pm0.43}$ |
| NAC | 45.42$_{\pm0.11}$ | 53.80$_{\pm0.08}$ | 65.83$_{\pm1.22}$ | 74.41$_{\pm0.35}$ | 60.79$_{\pm0.23}$ | 49.61$_{\pm0.06}$ | 67.01$_{\pm0.53}$ | 60.05$_{\pm0.58}$ |
| **KAN** | 58.37$_{\pm0.47}$ | 61.10$_{\pm0.53}$ | **84.13**$_{\pm0.35}$ | **83.30**$_{\pm0.35}$ | 70.40$_{\pm0.26}$ | 59.74$_{\pm0.46}$ | **79.28**$_{\pm0.18}$ | **71.46**$_{\pm0.40}$ |
| **ImageNet-1K FS Benchmark** | | | | | | | | |
| OpenMax | 53.79 | 60.28 | 80.30 | 73.54 | 71.88 | 57.03 | 75.24 | 67.96 |
| MSP | 56.66 | 64.93 | 76.35 | 69.33 | 71.28 | 60.79 | 72.32 | 67.71 |
| TempScale | 55.71 | 64.60 | 77.33 | 70.84 | 72.32 | 60.16 | 73.50 | 68.16 |
| ODIN | 54.22 | 60.59 | 77.43 | 76.04 | 73.40 | 57.41 | 75.62 | 68.34 |
| MDS | 39.22 | 52.83 | 54.06 | 86.26 | 60.75 | 46.02 | 67.02 | 58.62 |
| MDSEns | 37.13 | 47.80 | 53.32 | 73.39 | 53.24 | 42.47 | 59.98 | 52.98 |
| RMDS | 56.61 | 67.50 | 73.48 | 74.25 | 72.13 | 62.06 | 73.29 | 68.79 |
| Gram | 51.93 | 60.63 | 71.36 | 84.83 | 69.40 | 56.28 | 75.20 | 67.63 |
| EBO | 52.93 | 60.28 | 74.01 | 73.89 | 72.22 | 56.61 | 73.37 | 66.67 |
| GradNorm | **61.33** | 64.06 | 87.62 | 85.99 | 76.85 | 62.70 | 83.49 | 75.17 |
| ReAct | 55.34 | 64.51 | 87.93 | 81.08 | 79.34 | 59.93 | 82.78 | 73.64 |
| MLS | 53.56 | 61.43 | 75.61 | 73.42 | 72.66 | 57.49 | 73.90 | 67.34 |
| KLM | 56.87 | 67.26 | 80.88 | 70.73 | 74.74 | 62.06 | 75.45 | 70.10 |
| VIM | 45.88 | 59.12 | 72.22 | 93.09 | 75.01 | 52.50 | 80.10 | 69.06 |
| KNN | 43.78 | 59.86 | 67.79 | 90.29 | 69.98 | 51.82 | 76.02 | 66.34 |
| DICE | 54.01 | 60.29 | 82.52 | 83.89 | 76.42 | 57.15 | 80.94 | 71.43 |
| RankFeat | 50.30 | 40.37 | 34.34 | 66.29 | 44.98 | 45.34 | 48.54 | 47.26 |
| ASH | 54.66 | 66.38 | 89.23 | 89.53 | 81.47 | 60.52 | 86.75 | 76.25 |
| SHE | 58.15 | 64.27 | 84.71 | 87.48 | 76.92 | 61.21 | 83.04 | 74.31 |
| GEN | 52.95 | 62.73 | 78.47 | 71.82 | 72.62 | 57.84 | 74.31 | 67.72 |
| NAC | 52.48 | 66.49 | 88.92 | 92.77 | 80.76 | 59.48 | 87.48 | 76.28 |
| **KAN** | 55.88 | **69.55** | **91.55** | **93.45** | **82.15** | **62.71** | **89.05** | **78.52** |

Table 19: Tab. Med. Caucasian Eth. as InD (AUROC metric).

| Method | eICU - Eth. |
|---|---|
| MDS | **58.5**±2.2 |
| RMDS | 51.6±1.5 |
| KNN | 55.8±1.9 |
| VIM | 57.3±2.3 |
| SHE | 50.5±1.7 |
| KLM | 51.6±2.1 |
| OpenMax | 48.7±0.8 |
| MSP | 48.4±1.0 |
| MLS | 49.1±1.1 |
| TempScale | 48.4±1.0 |
| ODIN | 48.4±1.0 |
| EBO | 49.1±1.1 |
| GRAM | 50.8±2.7 |
| GradNorm | 50.4±3.4 |
| ReAct | 48.1±1.1 |
| DICE | 49.9±2.8 |
| ASH | 52.6±2.7 |
| **KAN** | **61.4**±3.1 |

Table 20: Tab. Med. > 70 y.o. as InD (AUROC metric).

| Method | eICU - Age |
|---|---|
| MDS | **50.8**±1.1 |
| RMDS | 48.3±0.7 |
| KNN | 49.6±0.2 |
| VIM | 48.8±0.1 |
| SHE | 50.4±0.7 |
| KLM | **51.0**±0.7 |
| OpenMax | 48.1±0.5 |
| MSP | 48.1±0.5 |
| MLS | 48.0±0.5 |
| TempScale | 48.1±0.5 |
| ODIN | 48.1±0.5 |
| EBO | 48.0±0.5 |
| GRAM | 48.6±0.6 |
| GradNorm | 48.7±0.8 |
| ReAct | 48.2±0.7 |
| DICE | 48.1±0.6 |
| ASH | 48.6±1.3 |
| **KAN** | **50.5**±0.5 |

Table 21: Tab. Med. Feature multiplication (AUROC metric).

| Method | eICU - Synthetic OOD | | | Avg Overall |
|---|---|---|---|---|
| | $\mathcal{F}=10$ | $\mathcal{F}=100$ | $\mathcal{F}=1000$ | |
| MDS | 59.9±1.4 | 79.5±1.4 | 87.5±0.9 | 75.63±1.26 |
| RMDS | 51.5±1.3 | 57.8±7.4 | 64.0±13.0 | 57.77±8.67 |
| KNN | 57.3±1.4 | 75.4±2.2 | 86.5±1.3 | 73.07±1.68 |
| VIM | 57.9±1.6 | 77.6±1.3 | **88.3**±0.7 | 74.60±1.16 |
| SHE | 55.7±1.3 | 71.2±2.9 | 80.4±1.6 | 69.10±2.05 |
| KLM | 54.1±0.8 | 63.1±1.1 | 72.1±4.2 | 63.10±2.55 |
| OpenMax | 51.0±0.9 | 56.1±2.7 | 71.4±3.2 | 59.50±2.45 |
| MSP | 50.9±0.6 | 55.8±2.7 | 71.3±3.1 | 59.33±2.40 |
| MLS | 50.9±0.7 | 55.8±3.1 | 70.8±3.1 | 59.17±2.56 |
| TempScale | 50.9±0.6 | 55.8±2.7 | 71.3±3.1 | 59.33±2.40 |
| ODIN | 50.9±0.6 | 55.9±2.7 | 71.4±3.1 | 59.40±2.40 |
| EBO | 50.9±0.7 | 55.5±3.1 | 70.2±3.2 | 58.87±2.60 |
| GRAM | 48.9±0.1 | 50.1±0.8 | 62.5±3.0 | 53.83±1.79 |
| GradNorm | 52.0±0.8 | 59.5±2.9 | 73.6±2.9 | 61.70±2.41 |
| ReAct | 50.9±0.8 | 55.7±3.4 | 69.9±3.2 | 58.83±2.73 |
| DICE | 52.2±1.5 | 62.8±2.5 | 75.6±1.4 | 63.53±1.87 |
| ASH | 50.1±1.4 | 54.6±2.1 | 69.7±1.3 | 58.13±1.64 |
| **KAN** | **64.6**±2.2 | **83.0**±2.6 | **89.8**±1.8 | **79.13**±2.22 |

Table 22: FPR@95 performance on CIFAR-10 and CIFAR-100 benchmarks.

| Method | Near OOD | | Far OOD | | | | Avg Near | Avg Far | Avg Overall |
|---|---|---|---|---|---|---|---|---|---|
| | CIFAR | TIN | MNIST | SVHN | Textures | Places365 | | | |
| **CIFAR-10 Benchmark** | | | | | | | | | |
| OpenMax | 48.06±3.25 | 39.18±1.44 | 23.33±4.67 | 25.40±1.47 | 31.50±4.05 | 38.52±2.27 | 43.62±2.27 | 29.69±1.21 | 34.33±3.11 |
| MSP | 53.08±4.86 | 43.27±3.00 | 23.64±5.81 | 25.82±1.64 | 34.96±4.64 | 42.47±3.81 | 48.17±3.92 | 31.72±1.84 | 37.21±4.19 |
| TempScale | 55.81±5.07 | 46.11±3.63 | 23.53±7.05 | 26.97±2.65 | 38.16±5.89 | 45.27±4.50 | 50.96±4.32 | 33.48±2.39 | 39.31±5.01 |
| ODIN | 77.00±5.74 | 75.38±6.42 | **23.83**±12.34 | 68.61±0.52 | 67.70±11.06 | 70.36±6.96 | 76.19±6.08 | 57.62±4.24 | 63.81±8.14 |
| MDS | 52.81±3.62 | 46.99±4.36 | 27.30±3.55 | 25.96±2.52 | 27.94±4.20 | 47.67±4.54 | 49.90±3.99 | 32.22±3.40 | 38.11±3.86 |
| MDSEns | 91.87±0.10 | 92.66±0.42 | **1.30**±0.51 | 74.34±1.04 | 76.07±0.17 | 94.16±0.33 | 92.26±0.20 | 61.47±0.48 | 71.73±0.53 |
| RMDS | 43.86±3.49 | 33.91±1.39 | 21.49±2.32 | 23.46±1.48 | 25.25±0.53 | 31.20±0.28 | 38.89±2.39 | 25.35±0.73 | 29.86±1.92 |
| Gram | 91.68±2.24 | 90.06±1.59 | 70.30±8.96 | 33.91±17.35 | 94.64±2.71 | 90.49±1.93 | 90.87±1.91 | 72.34±6.73 | 78.51±8.16 |
| EBO | 66.60±4.46 | 56.08±4.83 | **24.99**±12.93 | 35.12±6.11 | 51.82±6.11 | 54.85±6.52 | 61.34±4.63 | 41.69±5.32 | 48.24±7.39 |
| OpenGAN | 94.84±3.83 | 94.11±4.21 | 79.54±19.71 | **75.27**±26.93 | 83.95±14.89 | 95.32±4.45 | 94.48±4.01 | 83.52±11.63 | 87.17±15.21 |
| GradNorm | 94.54±1.11 | 94.89±0.60 | 85.41±4.85 | 91.65±2.42 | 98.09±0.49 | 92.46±2.28 | 94.72±0.82 | 91.90±2.23 | 92.84±2.46 |
| ReAct | 67.40±7.34 | 59.71±7.31 | **33.77**±18.00 | **50.23**±15.98 | 51.42±11.42 | 44.20±3.35 | 63.56±7.33 | 44.90±8.37 | **51.12**±11.75 |
| MLS | 66.59±4.44 | 56.06±4.82 | **25.06**±12.87 | 35.09±6.09 | 51.73±6.13 | 54.84±6.51 | 61.32±4.62 | 41.68±5.27 | 48.23±7.37 |
| KLM | 90.55±5.83 | 85.18±7.60 | 76.22±12.09 | 59.47±7.06 | 81.95±9.95 | 95.58±2.12 | 87.86±6.37 | 78.31±4.84 | 81.49±8.08 |
| VIM | 49.19±3.15 | 40.49±1.55 | 18.36±1.42 | **19.29**±0.41 | 21.14±1.83 | 41.43±2.17 | 44.84±2.31 | 25.05±0.52 | 31.65±1.94 |
| KNN | 37.64±0.31 | 30.37±0.65 | 20.05±1.36 | 22.60±1.26 | 24.06±0.55 | 34.01±0.38 | 34.01±0.38 | 24.27±0.40 | 27.52±0.88 |
| DICE | 73.71±7.67 | 66.37±7.68 | 30.83±10.54 | 36.61±4.74 | 62.42±4.79 | 77.19±12.60 | 70.04±7.64 | 51.76±4.42 | 57.85±8.50 |
| RankFeat | 65.32±3.48 | 56.44±5.76 | 61.86±12.78 | 64.49±7.38 | 59.71±9.79 | **43.70**±7.39 | 60.88±4.60 | 57.44±7.99 | 58.59±8.30 |
| ASH | 87.31±2.06 | 86.25±1.58 | 70.00±10.56 | 83.64±6.48 | 84.59±1.74 | 77.89±7.28 | 86.78±1.82 | 79.03±4.22 | 81.61±6.00 |
| SHE | 81.00±3.42 | 78.30±3.52 | **42.22**±20.59 | 62.74±4.01 | 84.60±5.32 | 76.36±5.32 | 79.65±3.47 | 66.48±5.98 | 70.87±9.31 |
| GEN | 58.75±3.97 | 48.59±2.34 | 23.00±7.75 | 28.14±2.59 | 40.74±6.61 | 47.03±3.22 | 53.67±3.14 | 34.73±1.58 | 41.04±4.87 |
| NAC | **35.15**±0.40 | **26.55**±0.18 | 15.13±2.62 | **14.33**±1.26 | **17.05**±0.60 | **26.73**±0.81 | **30.85**±0.19 | **18.31**±0.93 | **22.49**±1.27 |
| **KAN** | 40.81±2.91 | 34.17±1.85 | 11.63±3.11 | 14.13±2.38 | 21.30±0.98 | 32.49±2.21 | 37.49±2.36 | 19.89±1.19 | 25.75±2.35 |
| **CIFAR-100 Benchmark** | | | | | | | | | |
| OpenMax | 60.17±0.97 | 52.99±0.51 | 53.82±4.74 | 53.20±1.78 | 56.12±1.91 | **54.85**±1.42 | 56.58±0.73 | 54.50±0.68 | **55.19**±2.33 |
| MSP | 58.91±0.93 | **50.70**±0.34 | 57.23±4.68 | 59.07±2.53 | 61.88±1.28 | 56.62±0.87 | **54.80**±0.33 | 58.70±1.06 | **57.40**±2.30 |
| TempScale | **58.72**±0.81 | **50.26**±0.16 | 56.05±4.61 | 57.71±2.68 | 61.56±1.43 | 56.46±0.94 | **54.49**±0.48 | 57.94±1.14 | 56.79±2.31 |
| ODIN | 60.64±0.56 | 55.19±0.57 | 45.94±3.29 | 67.41±3.88 | 62.37±2.96 | 59.71±0.92 | 57.91±0.51 | 58.86±0.79 | 58.54±2.45 |
| MDS | 88.00±0.49 | 79.05±1.22 | 71.72±2.94 | 67.21±6.09 | 70.49±2.48 | 79.61±0.34 | 83.53±0.60 | 72.26±1.56 | 76.01±2.99 |
| MDSEns | 95.94±0.16 | 95.82±0.12 | **2.83**±0.86 | 82.57±2.58 | 84.94±0.83 | 96.61±0.17 | 95.88±0.04 | 66.74±1.04 | 76.45±1.17 |
| RMDS | 61.37±0.24 | **49.56**±0.90 | 52.05±6.28 | 51.65±3.68 | 53.99±1.06 | **53.57**±0.43 | 55.46±0.41 | 52.81±0.63 | **53.70**±3.03 |
| Gram | 92.71±0.64 | 91.85±0.86 | 53.53±7.45 | **20.06**±1.96 | 89.51±2.54 | 94.67±0.60 | 92.28±0.29 | 64.44±2.37 | 73.72±3.35 |
| EBO | **59.21**±0.75 | 52.03±0.50 | 52.62±3.83 | 53.62±3.14 | 62.35±2.06 | 57.75±0.86 | **55.62**±0.61 | 56.59±1.38 | 56.26±2.25 |
| OpenGAN | 78.83±3.94 | 74.21±1.25 | 63.09±23.25 | 70.35±2.06 | 74.77±1.78 | **73.75**±8.32 | 76.52±2.59 | 70.49±7.38 | **72.50**±10.28 |
| GradNorm | 84.30±0.36 | 86.85±0.62 | 86.97±1.44 | 69.90±7.94 | 92.51±0.61 | 85.32±0.44 | 85.58±0.46 | 83.68±1.92 | 84.31±3.32 |
| ReAct | 61.30±0.43 | 51.47±0.47 | 56.04±5.66 | 50.41±2.02 | 55.04±0.82 | 55.30±0.41 | 56.39±0.34 | 54.20±1.56 | **54.93**±2.50 |
| MLS | **59.11**±0.64 | 51.83±0.70 | 52.95±3.82 | 53.90±3.04 | 62.39±2.13 | 57.68±0.91 | **55.47**±0.66 | 56.73±1.33 | 56.31±2.24 |
| KLM | 84.77±2.95 | 71.07±0.59 | 73.09±6.67 | 50.30±7.04 | 81.80±5.80 | 81.40±1.58 | 77.92±1.31 | 71.65±2.01 | 73.74±4.82 |
| VIM | 70.59±0.43 | 54.66±0.42 | 48.32±1.07 | 46.22±5.46 | 46.86±2.29 | 61.57±0.77 | 62.63±0.27 | 50.74±1.00 | **54.70**±2.49 |
| KNN | 72.80±0.44 | **49.65**±0.37 | 48.58±4.67 | 51.75±3.12 | 53.56±2.32 | 60.70±1.03 | 61.22±0.14 | 53.65±0.28 | 56.17±2.53 |
| DICE | **60.98**±1.10 | 54.93±0.53 | 51.79±3.67 | 49.58±3.32 | 64.23±1.65 | 59.39±1.25 | 57.95±0.53 | 56.25±0.60 | 56.82±2.25 |
| RankFeat | 82.78±1.56 | 78.40±0.95 | 75.01±5.83 | 58.49±2.30 | 66.87±3.80 | 77.42±1.96 | 80.59±1.10 | 69.45±1.01 | 73.16±3.19 |
| ASH | 68.06±0.44 | 63.35±0.90 | 66.58±3.88 | 46.00±2.67 | 61.27±2.74 | 62.95±0.99 | 65.71±0.24 | 59.20±2.46 | 61.37±2.30 |
| SHE | 60.41±0.51 | 57.74±0.73 | 58.78±2.70 | 59.15±7.61 | 73.29±3.22 | 65.24±0.98 | 59.07±0.25 | 64.12±2.70 | 62.44±3.59 |
| GEN | **58.87**±0.69 | **49.98**±0.05 | 53.92±5.71 | 55.45±2.76 | 61.23±1.40 | 56.25±1.01 | **54.42**±0.33 | 56.71±1.59 | **55.95**±2.70 |
| NAC | 81.37±1.27 | 61.59±0.78 | 21.97±6.62 | **24.39**±4.46 | 40.65±1.94 | 73.57±1.16 | 71.48±1.03 | **40.14**±1.86 | 50.59±3.44 |
| **KAN** | 82.28±1.76 | 55.74±1.94 | 39.61±7.67 | 57.13±12.74 | 45.76±2.87 | 65.33±0.61 | 69.01±1.85 | 51.96±3.31 | **57.64**±6.28 |

Table 23: FPR@95 performance on ImageNet-200 FS and ImageNet-1K FS benchmarks.

| Method | Near OOD | | Far OOD | | | Avg Near | Avg Far | Avg Overall |
|---|---|---|---|---|---|---|---|---|
| | SSB-hard | NINCO | iNaturalist | Textures | OpenImage-O | | | |
| **ImageNet-200 FS Benchmark** | | | | | | | | |
| OpenMax | $91.55_{\pm0.05}$ | $85.00_{\pm0.26}$ | $68.19_{\pm0.77}$ | $76.72_{\pm0.33}$ | $77.39_{\pm0.28}$ | $88.27_{\pm0.14}$ | $74.10_{\pm0.46}$ | $79.77_{\pm0.41}$ |
| MSP | $89.08_{\pm0.03}$ | $\mathbf{79.30}_{\pm0.42}$ | $67.24_{\pm0.64}$ | $79.79_{\pm0.36}$ | $74.22_{\pm0.10}$ | $\mathbf{84.19}_{\pm0.21}$ | $73.75_{\pm0.26}$ | $77.93_{\pm0.38}$ |
| TempScale | $89.33_{\pm0.04}$ | $79.51_{\pm0.37}$ | $\mathbf{66.61}_{\pm0.68}$ | $79.67_{\pm0.39}$ | $74.10_{\pm0.22}$ | $\mathbf{84.42}_{\pm0.18}$ | $73.46_{\pm0.24}$ | $77.84_{\pm0.40}$ |
| ODIN | $91.71_{\pm0.09}$ | $86.66_{\pm0.39}$ | $\mathbf{65.50}_{\pm1.53}$ | $79.13_{\pm0.77}$ | $80.03_{\pm0.28}$ | $89.18_{\pm0.16}$ | $73.55_{\pm0.63}$ | $79.81_{\pm0.80}$ |
| MDS | $93.90_{\pm0.16}$ | $88.52_{\pm0.16}$ | $74.46_{\pm1.37}$ | $74.10_{\pm1.07}$ | $83.68_{\pm0.52}$ | $91.21_{\pm0.07}$ | $77.41_{\pm0.78}$ | $82.93_{\pm0.82}$ |
| MDSEns | $96.28_{\pm0.11}$ | $95.83_{\pm0.15}$ | $91.39_{\pm0.26}$ | $84.83_{\pm0.16}$ | $93.52_{\pm0.13}$ | $96.06_{\pm0.13}$ | $89.91_{\pm0.10}$ | $92.37_{\pm0.17}$ |
| RMDS | $89.44_{\pm0.17}$ | $79.23_{\pm0.48}$ | $65.62_{\pm1.16}$ | $76.73_{\pm0.68}$ | $74.82_{\pm0.09}$ | $84.33_{\pm0.31}$ | $72.39_{\pm0.64}$ | $77.17_{\pm0.64}$ |
| Gram | $\mathbf{86.37}_{\pm0.25}$ | $87.59_{\pm1.45}$ | $86.24_{\pm0.49}$ | $82.76_{\pm0.49}$ | $87.23_{\pm0.84}$ | $86.98_{\pm0.77}$ | $85.41_{\pm0.28}$ | $86.04_{\pm0.82}$ |
| EBO | $90.71_{\pm0.15}$ | $83.61_{\pm0.39}$ | $70.53_{\pm1.68}$ | $79.46_{\pm0.88}$ | $77.14_{\pm0.64}$ | $87.16_{\pm0.27}$ | $75.71_{\pm0.83}$ | $80.29_{\pm0.91}$ |
| OpenGAN | $95.90_{\pm0.21}$ | $94.47_{\pm1.39}$ | $81.85_{\pm4.09}$ | $84.76_{\pm1.87}$ | $86.71_{\pm0.61}$ | $95.18_{\pm0.60}$ | $84.44_{\pm0.80}$ | $88.74_{\pm2.12}$ |
| GradNorm | $91.16_{\pm0.41}$ | $91.68_{\pm0.31}$ | $78.68_{\pm2.25}$ | $82.11_{\pm1.95}$ | $84.81_{\pm0.53}$ | $91.42_{\pm0.32}$ | $81.86_{\pm0.44}$ | $85.69_{\pm1.37}$ |
| ReAct | $91.22_{\pm0.50}$ | $84.61_{\pm0.72}$ | $\mathbf{67.52}_{\pm2.93}$ | $72.82_{\pm1.56}$ | $74.81_{\pm0.18}$ | $87.91_{\pm0.61}$ | $71.72_{\pm1.54}$ | $\mathbf{78.20}_{\pm1.54}$ |
| MLS | $90.68_{\pm0.17}$ | $83.27_{\pm0.42}$ | $69.50_{\pm1.61}$ | $79.39_{\pm0.85}$ | $76.55_{\pm0.43}$ | $86.98_{\pm0.28}$ | $75.15_{\pm0.78}$ | $79.88_{\pm0.86}$ |
| KLM | $91.97_{\pm0.85}$ | $85.33_{\pm1.20}$ | $66.87_{\pm1.12}$ | $80.11_{\pm0.48}$ | $77.94_{\pm0.45}$ | $88.65_{\pm0.36}$ | $74.97_{\pm0.54}$ | $80.44_{\pm0.88}$ |
| VIM | $91.61_{\pm0.15}$ | $82.35_{\pm0.44}$ | $68.15_{\pm0.74}$ | $\mathbf{58.50}_{\pm0.85}$ | $74.54_{\pm0.51}$ | $86.98_{\pm0.28}$ | $\mathbf{67.06}_{\pm0.63}$ | $75.03_{\pm0.59}$ |
| KNN | $91.73_{\pm0.15}$ | $81.23_{\pm0.35}$ | $69.10_{\pm0.74}$ | $69.06_{\pm0.22}$ | $74.43_{\pm0.66}$ | $86.48_{\pm0.25}$ | $70.86_{\pm0.50}$ | $77.11_{\pm0.49}$ |
| DICE | $90.94_{\pm0.09}$ | $84.24_{\pm0.51}$ | $72.10_{\pm1.81}$ | $78.84_{\pm0.87}$ | $77.79_{\pm0.68}$ | $87.59_{\pm0.29}$ | $76.24_{\pm0.78}$ | $80.78_{\pm0.98}$ |
| RankFeat | $95.78_{\pm0.10}$ | $96.98_{\pm0.18}$ | $99.16_{\pm0.31}$ | $99.75_{\pm0.28}$ | $98.16_{\pm0.35}$ | $96.38_{\pm0.14}$ | $99.02_{\pm0.30}$ | $97.97_{\pm0.26}$ |
| ASH | $90.29_{\pm0.42}$ | $84.21_{\pm0.45}$ | $63.16_{\pm1.71}$ | $65.99_{\pm0.56}$ | $\mathbf{71.69}_{\pm0.41}$ | $87.25_{\pm0.40}$ | $66.95_{\pm0.64}$ | $75.07_{\pm0.87}$ |
| SHE | $91.16_{\pm0.13}$ | $86.49_{\pm0.62}$ | $71.48_{\pm2.66}$ | $78.98_{\pm1.32}$ | $79.50_{\pm0.69}$ | $88.82_{\pm0.36}$ | $76.65_{\pm0.88}$ | $81.52_{\pm1.39}$ |
| GEN | $89.59_{\pm0.05}$ | $\mathbf{80.12}_{\pm0.33}$ | $66.47_{\pm0.74}$ | $79.30_{\pm0.45}$ | $73.96_{\pm0.26}$ | $84.85_{\pm0.14}$ | $73.24_{\pm0.40}$ | $77.89_{\pm0.43}$ |
| NAC | $92.75_{\pm0.35}$ | $88.83_{\pm0.12}$ | $72.57_{\pm1.49}$ | $69.08_{\pm0.63}$ | $79.55_{\pm0.72}$ | $90.79_{\pm0.16}$ | $73.73_{\pm0.66}$ | $80.56_{\pm0.81}$ |
| KAN | $94.67_{\pm1.20}$ | $87.35_{\pm1.49}$ | $\mathbf{64.89}_{\pm3.67}$ | $68.75_{\pm1.79}$ | $78.84_{\pm1.55}$ | $91.01_{\pm1.35}$ | $70.83_{\pm0.25}$ | $\mathbf{78.90}_{\pm2.13}$ |
| **ImageNet-1K FS Benchmark** | | | | | | | | |
| OpenMax | 86.76 | 76.41 | 49.98 | 62.28 | 60.05 | 81.59 | 57.44 | 67.10 |
| MSP | 84.47 | 72.37 | 61.75 | 75.27 | 67.29 | 78.42 | 68.10 | 72.23 |
| TempScale | **84.23** | 71.55 | 58.16 | 72.87 | 64.35 | 77.89 | 65.13 | 70.23 |
| ODIN | 86.01 | 80.40 | 57.10 | 67.34 | 65.45 | 83.21 | 63.29 | 71.26 |
| MDS | 95.41 | 86.89 | 83.38 | 55.95 | 82.16 | 91.15 | 73.83 | 80.76 |
| MDSEns | 96.22 | 93.56 | 87.47 | 78.56 | 92.72 | 94.89 | 86.25 | 89.71 |
| RMDS | 87.09 | **70.28** | 55.68 | 67.83 | 61.30 | 78.69 | 61.61 | 68.44 |
| Gram | 90.87 | 86.24 | 72.69 | 64.65 | 79.05 | 88.55 | 72.13 | 78.70 |
| EBO | 86.19 | 76.15 | 55.18 | 66.05 | 60.55 | 81.17 | 60.59 | 68.82 |
| GradNorm | 84.44 | 85.45 | 45.63 | 55.84 | 76.76 | 84.95 | 59.41 | 69.62 |
| ReAct | 86.90 | 72.97 | 40.64 | 53.47 | 55.89 | 79.94 | 50.00 | 61.97 |
| MLS | 85.98 | 75.43 | 54.59 | 66.34 | 60.39 | 80.71 | 60.44 | 68.55 |
| KLM | 90.19 | 74.14 | 58.60 | 68.69 | 66.22 | 82.17 | 64.50 | 71.57 |
| VIM | 88.85 | 77.85 | 55.59 | 33.64 | 57.23 | 83.35 | 48.82 | 62.63 |
| KNN | 90.50 | 74.77 | 62.46 | 42.59 | 65.03 | 82.64 | 56.69 | 67.07 |
| DICE | 86.93 | 79.87 | 54.89 | 63.84 | 66.53 | 83.40 | 61.75 | 70.41 |
| RankFeat | 93.17 | 96.21 | 96.43 | 83.49 | 93.59 | 94.69 | 91.17 | 92.58 |
| ASH | 84.49 | 70.74 | **36.82** | 38.34 | **52.33** | **77.61** | 42.50 | **56.54** |
| SHE | 85.10 | 80.47 | 50.97 | 52.17 | 69.38 | 82.78 | 57.50 | 67.62 |
| GEN | 85.72 | 72.38 | 50.61 | 66.40 | 57.67 | 79.05 | 58.23 | 66.56 |
| NAC | 89.54 | 76.53 | 36.88 | **29.19** | 58.48 | 83.03 | **41.52** | 58.12 |
| KAN | 89.64 | 74.57 | 42.67 | 37.79 | 67.66 | 82.11 | 49.37 | 62.47 |

Table 24: Tab. Med. Caucasian Eth. as InD (FPR@95 metric).

| Method | eICU - Eth. |
|---|---|
| MDS | $\mathbf{89.7}_{\pm4.3}$ |
| RMDS | $\mathbf{93.1}_{\pm1.6}$ |
| KNN | $\mathbf{91.8}_{\pm2.0}$ |
| VIM | $\mathbf{90.7}_{\pm4.3}$ |
| SHE | $\mathbf{93.2}_{\pm1.8}$ |
| KLM | $\mathbf{92.8}_{\pm1.4}$ |
| OpenMax | $\mathbf{93.8}_{\pm1.9}$ |
| MSP | $\mathbf{93.5}_{\pm1.6}$ |
| MLS | $\mathbf{93.4}_{\pm1.6}$ |
| TempScale | $\mathbf{93.5}_{\pm1.6}$ |
| ODIN | $\mathbf{93.5}_{\pm1.6}$ |
| EBO | $\mathbf{93.3}_{\pm1.6}$ |
| GRAM | $\mathbf{93.6}_{\pm2.3}$ |
| GradNorm | $\mathbf{92.0}_{\pm3.2}$ |
| ReAct | $\mathbf{94.5}_{\pm2.3}$ |
| DICE | $\mathbf{93.0}_{\pm2.2}$ |
| ASH | $\mathbf{92.6}_{\pm2.1}$ |
| **KAN** | $91.6_{\pm3.4}$ |

Table 25: Tab. Med. > 70 y.o. as InD (FPR@95 metric).

| Method | eICU - Age |
|---|---|
| MDS | $\mathbf{94.3}_{\pm0.5}$ |
| RMDS | $95.6_{\pm0.1}$ |
| KNN | $\mathbf{94.7}_{\pm0.2}$ |
| VIM | $\mathbf{94.6}_{\pm0.0}$ |
| SHE | $95.1_{\pm0.3}$ |
| KLM | $95.2_{\pm0.3}$ |
| OpenMax | $95.2_{\pm0.1}$ |
| MSP | $95.3_{\pm0.2}$ |
| MLS | $95.3_{\pm0.1}$ |
| TempScale | $95.3_{\pm0.2}$ |
| ODIN | $95.3_{\pm0.2}$ |
| EBO | $95.2_{\pm0.1}$ |
| GRAM | $95.0_{\pm0.6}$ |
| GradNorm | $95.3_{\pm0.2}$ |
| ReAct | $95.2_{\pm0.2}$ |
| DICE | $\mathbf{94.9}_{\pm0.1}$ |
| ASH | $95.3_{\pm0.1}$ |
| **KAN** | $97.3_{\pm0.9}$ |

Table 26: Tab. Med. Feature multiplication (FPR@95 metric).

| Method | eICU - Synthetic OOD | | | Avg Overall |
|---|---|---|---|---|
| | $\mathcal{F}=10$ | $\mathcal{F}=100$ | $\mathcal{F}=1000$ | |
| MDS | $81.6_{\pm1.3}$ | $\mathbf{48.1}_{\pm4.1}$ | $28.0_{\pm4.9}$ | $52.57_{\pm3.76}$ |
| RMDS | $91.9_{\pm2.3}$ | $78.7_{\pm10.4}$ | $64.2_{\pm17.7}$ | $78.27_{\pm11.93}$ |
| KNN | $83.6_{\pm1.8}$ | $53.4_{\pm5.3}$ | $29.8_{\pm5.6}$ | $55.60_{\pm4.57}$ |
| VIM | $81.9_{\pm2.8}$ | $48.7_{\pm3.5}$ | $\mathbf{26.6}_{\pm3.0}$ | $\mathbf{52.40}_{\pm3.11}$ |
| SHE | $88.2_{\pm2.5}$ | $63.0_{\pm6.6}$ | $41.6_{\pm4.4}$ | $64.27_{\pm4.80}$ |
| KLM | $91.0_{\pm1.8}$ | $79.5_{\pm4.4}$ | $60.9_{\pm9.7}$ | $77.13_{\pm6.24}$ |
| OpenMax | $91.2_{\pm1.9}$ | $80.7_{\pm4.8}$ | $60.6_{\pm9.2}$ | $77.50_{\pm6.09}$ |
| MSP | $91.0_{\pm1.4}$ | $79.2_{\pm2.5}$ | $58.8_{\pm5.3}$ | $76.33_{\pm3.48}$ |
| MLS | $91.2_{\pm1.0}$ | $80.4_{\pm3.7}$ | $61.0_{\pm7.7}$ | $77.53_{\pm4.97}$ |
| TempScale | $91.0_{\pm1.4}$ | $79.2_{\pm2.5}$ | $58.8_{\pm5.3}$ | $76.33_{\pm3.48}$ |
| ODIN | $90.9_{\pm1.4}$ | $79.0_{\pm2.4}$ | $58.4_{\pm5.1}$ | $76.10_{\pm3.35}$ |
| EBO | $91.6_{\pm0.8}$ | $82.0_{\pm5.1}$ | $63.7_{\pm9.0}$ | $79.10_{\pm5.99}$ |
| GRAM | $94.1_{\pm0.6}$ | $89.8_{\pm4.7}$ | $79.9_{\pm8.3}$ | $87.93_{\pm5.52}$ |
| GradNorm | $91.3_{\pm1.4}$ | $78.6_{\pm4.5}$ | $58.3_{\pm6.7}$ | $76.07_{\pm4.73}$ |
| ReAct | $91.8_{\pm0.7}$ | $82.9_{\pm3.9}$ | $65.5_{\pm6.7}$ | $80.07_{\pm4.49}$ |
| DICE | $88.3_{\pm1.1}$ | $66.9_{\pm2.3}$ | $45.1_{\pm2.7}$ | $66.77_{\pm2.14}$ |
| ASH | $91.6_{\pm0.7}$ | $81.0_{\pm4.5}$ | $61.0_{\pm6.9}$ | $77.87_{\pm4.77}$ |
| **KAN** | $\mathbf{77.2}_{\pm4.0}$ | $49.3_{\pm5.5}$ | $33.7_{\pm5.2}$ | $53.42_{\pm4.94}$ |

