# OpenReview forum: "Advancing Out-of-Distribution Detection via Local Neuroplasticity"
_ICLR.cc/2025/Conference — ICLR 2025 Poster_

### Official Review · Reviewer_WMKe · 2024-10-29

**Soundness:** 3
**Presentation:** 3
**Contribution:** 2
**Rating:** 5
**Confidence:** 3

**Summary:**

The authors propose to use Kolmogorov-Arnold Networks (KAN) for out-of-distribution detection. The key advantage of KANs is their plasticity which results in avoiding catastrophic forgetting. The authors show that this property can be leveraged to detect OOD samples.

The method demonstrates good performance on small datasets, but the proposed method does not properly address the shortcomings of the KAN architecture, and the method was not validated in terms of scalability to realistic problems. Overall I rate weak reject.

**Strengths:**

- Originality: Given that KANs are a novel type of architecture the research is a very current
- The method is evaluated on image and tabular data, demonstrating feasibility across different domains.
- Performance: The performance on the benchmarks is convincing and demonstrates superiority over a vast set of previous methods
- Exhaustive experimentation on toy datasets including multiple important ablations that erase questions (such as stochasticity)

**Weaknesses:**

Major:
- Scalability: No experiments demonstrate the method's scalability to larger images or real-world problems.
- Insufficient capturing of joint distribution: I believe the partitioning problem of KANs is very severe. While the problem is mentioned I believe it is not properly addressed. Essentially, by partitioning the dataset you are just scaling the problem down to subclasses. What if the l-shaped differences, that you mention in Table 2, appear on an intra-class level instead of a class level? While this may work for toy data if the data is sufficiently separable using k-means or class labels directly, I doubt it will work for more difficult problems such as MVTech.
- The influence of Model capacity is unclear: KANs are known for their improvements in lack of catastrophic forgetting. How does the model size influence this. Additionally, if KANs treat features individually, the difficulty of the problem and the necessary capacity of the method scales drastically with the image size.

**Questions:**

Line 43 has wrong citation

You mention that the hyperpareter search can be quite challenging. How did you decide for the parameter space especially regarding number of epochs, learning rate, partitionings?

---

> ### Author Response · Authors · 2024-11-25
> **Response to Reviewer WMKe**
>
> We thank the reviewer for their detailed feedback and constructive criticism. We appreciate the acknowledgment of the originality, performance, and exhaustive experimentation of our method. Below, we address the concerns raised in the review:
>
> **Q1**: No experiments demonstrate the method's scalability to larger images or real-world problems.\
> **A1**: We acknowledge the absence of large-scale experiments in our initial submission. To address this, we have conducted additional experiments on the ImageNet-200 dataset, which is part of the OpenOOD benchmarks and contains five times more images that are seven times larger compared to the CIFAR benchmark. We specifically considered the full-spectrum version of the benchmark as it makes the detection problem more challenging and closer to real-world situations by adding various covariate-shifted samples to the InD test set. As shown in Table 2 of Section 3.2 of our revised manuscript, the KAN detector ranks first, surpassing the previous best method by approximately 4%.
>
> **Q2**: The partitioning problem of KANs is very severe.\
> **A2**: As suggested by Reviewer Asxo, we incorporated additional experiments on regression datasets where no classes exist. We show that even when classes are not available, the partitioning method still performs well (see Appendix A.2). We hope that this experiment, along with the tests on the large-scale ImageNet-200, demonstrates that the partitioning method is an essential component of our detector and that it effectively works in various scenarios.
>
> **Q3**: The influence of model capacity is unclear.\
> **A3**: The model size in our case is controlled by three factors: the input size, the output size, and the grid size. The first two are generally dictated by the problem itself. Note, however, that the input to our detector is not the raw image but the latent features space of the backbone. Typically, the latent space has a size drastically smaller compared to the input image, ensuring that larger images do not compromise the scalability of our detector. The effect on performance given by the grid size, and thus the model size, is shown in Table 8 of our manuscript. We also added this explanation to the revised manuscript in Appendix A.11.
>
> **Q4**: Line 43 has an incorrect citation.\
> **A4**: Thank you for pointing this out. We have corrected the citation in Line 43.
>
> **Q5**: Clarification on how the hyperparameter search is conducted.\
> **A5**: We tried to keep the search space quite large to ensure that we capture the optimal values. For the parameters related to the KAN (i.e. grid size, learning rate, epochs) we used similar ranges to what described in the examples of the original KAN paper. We have added the ranges considered for each parameter in Appendix A.12.
>
> Thank you once again for your valuable feedback. We have carefully addressed all the comments and made revisions based on your suggestions, which we believe have greatly enhanced the quality and clarity of our paper.

---

> > ### Author Response · Authors · 2024-11-30
> >
> > We carried out further experiments on the OpenOOD ImageNet-1K (full-spectrum) benchmark.
> > Our method now holds first place on this leaderboard, outperforming the previous best by 2%.
> > Detailed results have been shared in the general comment "Evaluation on ImageNet-1k" for your convenience.
> >
> > We hope you will take these additional results into account during your evaluation.\
> > Thank you again for your valuable feedback.

---

### Official Review · Reviewer_cRaW · 2024-11-03

**Soundness:** 3
**Presentation:** 3
**Contribution:** 3
**Rating:** 6
**Confidence:** 4

**Summary:**

Authors utilize Kolmogorov Arnold Networks (KAN) for out of
distribution detection. The main idea is to leverage the fact that KAN
uses BSplines as non-linear functions. Feature values that appear
within InD, if they are concentrated in certain part of the feature
space - which is $\mathbb{R}$ in this case, will only modify certain
BSpline coefficients. In this scenario when a feature value that is
different than the InD comes, the BSpline coefficients at those
locations will not have been modified during training. Hence, the
difference in activation between trained and untrained network will be
low. Experiments with benchmark datasets and comparisons with large
set of alternatives are presented.

**Strengths:**

+ The topic is very relevant.
+ The idea is novel and quite intuitive.
+ The results are motivating. Even though this is not the best
  performing all around, it is one of the top algorithms.
+ Authors do a great job explaining the method as well as motivating
  the approach.
+ Large set of experiments.

**Weaknesses:**

- The model - due to KANs - is heavily univariate. While authors do
  dataset partitioning to alleviate the problem, I do not see how they
  can actually do so. Unsupervised combinations of features are
  mentioned, however, their applicability also raises questions.
- Partitioning the dataset requires having multiple trained models,
  which limits the applicability of the approach for large scale
  problems.
- KANs are interesting but most recent work do not use these
  networks. This naturally limits the applicability of the approach.

**Questions:**

- It is not clear how different KAN$_i$'s are trained. It would be
  good to explain this a bit more in depth.
- Authors state that the method can be seamlessly integrated with any
  pre-trained model. I do not really understand this. Doesn't one need
  to use KAN model for this?
- How are the pre-trained backbones used for KAN? Does one use the
  features extracted from these networks and build classifiers and
  regressors with KAN architecture?
- Authors state that hyperparameters are tuned using a validation
  set. How much do the trained hyperparameters generalize to OOD types
  unseen in the validation set?

---

> ### Author Response · Authors · 2024-11-25
> **Response to Reviewer cRaW**
>
> Thank you for your thorough review and constructive feedback on our submission. We appreciate your positive remarks regarding the relevance, novelty, and motivation of our work, as well as your recognition of the extensive set of experiments we conducted. Below, we address the specific questions and concerns you raised:
>
> **Q1**: Univariate Nature of KANs\
> **A1**: We acknowledge that KANs, being inherently univariate, might seem to limit their applicability. However, our dataset partitioning strategy—whether by class labels or clustering methods such as k-means—enables us to divide complex and correlated feature distributions into smaller ones that can be well-approximated using only the marginal feature distribution. Consequently, the KAN detector can effectively process these partitions. We have clarified this point in Appendix A.7 together with new experiments and discussions on alternative clustering techniques. As suggested by another reviewer, we also tested the partitioning method of our detector in regression-based datasets where no classes exist and showed that the partitioning method still performs well (see Appendix A.2).
>
> **Q2**: Scalability to Large Datasets\
> **A2**: We demonstrate that our method effectively handles large-scale datasets with a new experiment on the ImageNet-200 benchmark which contains five times more images that are seven times larger compared to the CIFAR benchmark. Here our method outperforms all previous baselines (see Table 2 of Section 3.2 for more details). We have also revised Appendix A.11 to include a discussion on the method's complexity and scalability.
>
> **Q3**: Recent Usage of KANs\
> **A3**: We agree that KANs are relatively new and not yet widely adopted. Our focus was on using the KAN detector as a post-hoc method. This means that it can be applied to any existing backbone (e.g., CNN or Transformer) without influencing the classification output of the backbone itself. We show that this works for small- and large-scale datasets, image and tabular data, and different backbone models.
>
> **Q4**: Training of Different KANs\
> **A4**: We apologize for the lack of clarity regarding the training process of different KANs. Each KAN is initialized identically, with the only difference being the data subset (partition) on which they are trained. The training task is the same for all models, and in our case, we used the same loss function as the backbone. We have clarified this point in lines 191-192 of our revised manuscript.
>
> **Q5**: Integration with Pre-trained Models\
> **A5**: To clarify, our approach (like other post-hoc methods) does not replace pre-trained models but rather complements them. The pre-trained model, such as ResNet-18, is used for feature extraction. These features are then processed by our KAN-based detector (or any other considered post-hoc technique) in a subsequent phase. These backbone models do not have to be based on KANs; they can follow any architecture, such as fully connected MLPs, ResNets, or Transformers.
>
> **Q6**: Use of Pre-trained Backbones\
> **A6**: The primary job of the backbones is to perform the classification or regression task. The OOD detector is applied afterward to detect semantically different samples (e.g., samples that do not belong to the training classes), which would yield incorrect predictions by the backbone. From the detector's perspective, the backbone's job is simply to provide the latent features. We clarified this in the revised manuscript at lines 266-268.
>
> **Q7**: Generalization of Hyperparameters\
> **A7**: The validation set contains both InD and OOD samples. However, the OOD samples encountered at test time are of a different type as they belong to different datasets and hence classes.
>     For instance, on the CIFAR-10 benchmark, the validation set includes only OOD samples of the "near" type (CIFAR-100 and TIN datasets) while the test set contains also four datasets with "far" OOD samples. Our method performs well on both categories indicating that the selected hyperparameters generalize well even when new OOD type are encountered.
>
>
> We greatly appreciate the insightful feedback provided. We have implemented the recommended changes and believe these revisions have substantially improved the paper's quality and clarity.

---

> > ### Author Response · Authors · 2024-11-30
> >
> > We performed additional tests on the OpenOOD ImageNet-1K (full-spectrum) benchmark.
> > Our method achieves first place on this leaderboard as well, exceeding the performance of the previously best method by 2%.
> > The detailed results are available in the general comment "Evaluation on ImageNet-1k" for your review.
> >
> > We kindly ask you to consider these findings in your evaluation process.\
> > Once again, we appreciated your constructive feedback.

---

### Official Review · Reviewer_oW4J · 2024-11-04

**Soundness:** 3
**Presentation:** 3
**Contribution:** 3
**Rating:** 5
**Confidence:** 4

**Summary:**

This paper introduces a novel OOD detection method that leverages the unique local neuroplasticity of Kolmogorov-Arnold Networks (KANs). By comparing the activation patterns of a trained KAN against its untrained counterpart, the method identifies OOD samples across diverse benchmarks, including computer vision and tabular medical data. Experimental results demonstrate that the KAN-based approach outperforms existing methods and shows resilience to variations in in-distribution dataset sizes. This robust, adaptable approach makes KANs a promising tool for enhancing the reliability of ML systems.

**Strengths:**

1. It introduces an innovative approach to OOD detection, offering fresh ideas and a unique viewpoint that advances the current understanding of OOD detection techniques.
2. The paper effectively harness the neuroplasticity characteristic of KANs, ensuring that learning new tasks only affects the network regions activated by the training data, effective motivation for OOD detection.
3. The paper includes thorough experiments on standard benchmarks.

**Weaknesses:**

1. While the core idea is clear, the method appears loosely structured. Specifically, the role of multiplying location-specific information with regions activated by InD samples to achieve the delta function (used in the score function) is unclear (e.g., Eqn 5). Additionally, no study is provided to analyze these aspects, leaving parts of the methodology unexplored.
2. The paper does not present or discuss the generalization performance of models when KANs are incorporated into the training scheme.
3. Results on CIFAR-100 indicate minimal advantage over existing methods, as the improvements in detection performance appear statistically insignificant.
4. Including a discussion on the computational cost of the proposed method would strengthen the paper. Given that the approach involves dividing the dataset into different groups, insights into computational efficiency would enhance understanding of the method’s practicality.

**Questions:**

Please answer the points raised in the questions.

---

> ### Author Response · Authors · 2024-11-25
> **Response to Reviewer oW4J**
>
> We sincerely appreciate the time and effort invested in providing valuable feedback on our submission. We are pleased that our contributions and thorough experiments have been acknowledged. Below, we address the specific points raised:
>
> **Q1**: Method structure and clarity (Eqn 5):\
> **A1**: We acknowledge the need for further clarity regarding the method structure. Intuitevely, many methods define the boundary surrounding the InD based on the training samples and classify samples at inference time based on their distance to this boundary. In our approach, the InD boundary is encapsulated within the spline trainable coefficients, while regions activated by InD samples are utilized to determine the distance from the boundary through the aggregation of the delta matrix. we clarified this in line 136-137 of the revised manuscript.
>
> **Q2**: Generalization performance with KANs:\
> **A2**: We agree that integrating KANs in the training scheme can influence the final performance. We have included new experiments in Appendix A.2 to illustrate that our detector can be applied directly to the data features without an additional backbone model. However, it is important to emphasize that our method functions as a post-hoc processor, applied to a pre-trained backbone. Consequently, training our detector does not impact the backbone model as it function as a separate block. The advantage of post-hoc methods is their ease of integration with different backbones without requiring additional training, even in scenarios where no feature-extractor backbone is available.
>
> **Q3**: Results on CIFAR-100:\
> **A3**: Although the improvements on CIFAR-100 are minimal, it is noteworthy that our method is either the best or statistically similar to the best-performing method across a wide range of benchmarks on the overall average AUROC metric. This consistency is not observed in other approaches. For instance, while MDS performs well on medical benchmarks, it shows poor performance on CIFAR-10 and CIFAR-100. Conversely KNN excels on CIFAR benchmarks but underperforms on the medical datasets.
> To highlight this aspect even more, we conducted additional experiments on the ImageNet-200 benchmark (Table 2 of Section 3.2) where the KAN outperforms the previous best method by approximately 4\%.
> This highlights the robustness and versatility of our method.
>
> **Q4**: Computational cost:\
> **A4**: We have extended Appendix A.11 to include not only inference time but also the setup time (which includes extracting the latent features from the backbone, the partitioning method and the trainings of the KANs). We reported the results as a function of the dataset size and the number fo partitions. The results shows that the setup time of our detector scales linearly with the dataset size, in-line with other methods.
>
> Thank you for your thoughtful comments. We have revised the manuscript in response to your suggestions, leading to significant enhancements in both quality and clarity.

---

> > ### Author Response · Authors · 2024-11-30
> >
> > We have conducted additional experiments on the OpenOOD ImageNet-1K (full-spectrum) benchmark. Our method ranks first also on this leaderboard, surpassing the previously best method by 2%. You can find detailed results in a general comment we have provided.
> >
> > We kindly ask you to consider also these new results in your evaluation.\
> > Thank you again for your insightful feedback.

---

### Official Review · Reviewer_Asxo · 2024-11-05

**Soundness:** 3
**Presentation:** 3
**Contribution:** 2
**Rating:** 5
**Confidence:** 4

**Summary:**

The paper introduces a new out-of-distribution (OOD) detection method leveraging Kolmogorov-Arnold Networks (KANs), which utilize “local neuroplasticity” to differentiate in-distribution (InD) data from OOD data via comparing the activation patterns of a trained KAN against an untrained counterpart. KANs stand out due to their spline-based architecture, which preserve specific network regions during training, aiding in the OOD detection.

**Strengths:**

The described method is clearly defined and is easy to reproduce.

The method is validated across image and tabular medical data benchmarks, demonstrating improved performance and robustness compared to other state-of-the-art OOD detectors.

The findings highlight KANs' potential in enhancing model reliability across diverse environments by maintaining high detection accuracy, even with a relatively small training dataset.

The results (although not on all datasets) look promising in terms of different OOD detection accuracy especially for the case of low number of training samples.

**Weaknesses:**

Despite the clarity, some steps of the approach implementation look like ad-hoc tricks for improving the method’s performance without developing a deep intuition why a particular step is better than alternatives (please, see questions below for details).

The fact that not all datasets (leaderboards) from the OpenOOD were used for testing the approach, along with the obtained not perfect results on CIFAR-100, suggest that the datasets were selected manually. The authors need to prove absence of any selection bias.

I am strongly concerned about the scalability of the proposed method, which requires splitting the training dataset into a number of subsets and fitting a model per a subset (see comments below).

The method resembles feature-preprocessor (backbone-dependent), being not applicable to the case where a good feature extractor is not known.

**Questions:**

Questions and suggestions:

Major:
Testing of approach on other large-scale datasets would be beneficial, consider other leaderboards from openOOD like ImageNet-200, 1K.
The choice of the K-means clustering approach looks quite arbitrary for initial data splitting. Why not use other clustering approaches like DBScan, Spectral, Agglomerative or even Gausian mixture? I believe, K-means choice should be justified here.

One can assume a dataset with a lot of natural clusters (like ImageNet-1K) will require a lot of time for training KANs. Show that the approach is actually scalable, robust, and not computationally burdensome in case of a large number of clusters.

The robustness of clustering approach is not evident for the case of regression task due to the poor internal separability of data clusters. I suggest adding one example of OOD detection where the training dataset is directly related to the regression task.

The method looks strongly backbone dependent and may be poorly working for the plethora of practical tasks where the good backbone feature extractor is not known. Is it possible to exemplify the method robustness for the case of the absence of backbone preprocessor?
Probably, some classic ML tabular datasets (e.g. from sklearn) could be useful here.

“Importantly, our experiments show that the previous methods suffer from a non-optimal InD dataset size” - this statement requires more experimental support. Currently, the method superiority was shown only for the CIFAR-10 dataset.

Minor:
Line 183 (figure caption): “- “(e) InD score S(x)∀x ∈ [−1,1] “ - why the InD score can take negative values? The original formula (5) contains absolute value operation brackets. Is this the typo?

Line 187: “A simple, yet effective approach is to split the dataset based on class labels.”  - It is not obvious how to train KANs in case of such splitting. One can imagine a situation where positive class is OOD for a KAN trained on samples of negative class, and the maximization scoring procedure identifies positive class as an OOD. This point should be clarified or rephrased.

I’m interested if the method will be robust for the case of NaN-enriched data samples? It is not a request for an additional analysis but rather an interesting point for the discussion of method limitations.

---

> ### Author Response · Authors · 2024-11-25
> **Reply to Reviewer Asxo**
>
> We sincerely thank the reviewer for the detailed feedback and the recognition of our method's clarity and reproducibility. We greatly appreciate the constructive criticisms and suggestions, which have been instrumental in refining our work. Below, we address each of the major and minor concerns raised.
>
> **Q1.1/Q2**: Testing on other large-scale datasets and scalability and computational burden: \
> **A1.1/A2**: We understand your concerns regarding scalability, which were also raised by other reviewers. To address this, we included experiments on the OpenOOD ImageNet-200 leaderboard, where our method ranks first with an overall average AUROC approximately 4\% higher than the previously best performing method (detailed results are available in Table 2 of Section 3.2 in the revised manuscript). Due to time constraints, we could not include all OpenOOD leaderboards and other suggested benchmarks. However, we believe that the ImageNet-200, with five times more samples than the CIFAR leaderboards and images seven times larger, effectively demonstrates our method's scalability. To further eliminate any selection bias, we opted for the full-spectrum version of the ImageNet-200 benchmark, which includes covariate-shifted InD samples, making the detection challenge more complex and closer to real-world applications. We also expanded Appendix A.11 with a detailed discussion on the method's complexity showing that the most impactful factor is the dataset size and that our method has a similar scaling law to other approaches such as KNN.
>
> **Q1.2**: Choice of K-means clustering:\
> **A1.2**: The choice of K-means was motivated by its simplicity and low computational overhead. Based on this review, we conducted additional experiments testing several alternative clustering methods and found that the choice of clustering method does not significantly affect detection performance. These results are now included in the revised manuscript in Appendix A.7.
>
> **Q3**: Regression task example:\
> **A3**: To demonstrate that our method performs well on regression tasks, we tested it on the California Housing and Wine Quality datasets and showed that our method outperfoms the KNN detector on both of them. To further validate that the partitioning method is a core component of our detector and not merely a performance-improvement trick, we also used the Friedman synthetic dataset. The results show that the partitioning method is effective even for regression tasks with complexly correlated input features. These experiments have been added to Appendix A.2 in the revised manuscript.
>
> **Q4**: Backbone dependency:\
> **A4**: As suggested, we performed OOD detection on datasets without using a backbone or any other feature-extraction method. We used the same three datasets mentioned in the above answer A3 (i.e., California Housing, Wine Quality, and Friedman synthetic dataset). Our method showed superior performance compared to the KNN baseline on all three datasets, proving its applicability even in the absence of a backbone. In Appendix A.2, we also clarified that our method does not require any additional information from the backbone other than the features, unlike NAC, which requires the gradient of the backbone network.
>
> **Q5**: Support for the statement on InD dataset size:\
> **A5**: To better support our claim, we repeated the same experiment on the CIFAR-100 benchmark, and the results show a similar conclusion (see Table 6 in the revised manuscript).
>
> **Q6**: InD score negative values (line 183):\
> **A6**: It is correct that the InD score cannot be negative. Here the range [-1, 1] here refers to the support of the input space \( x \).
>
> **Q7**: Clarification on dataset splitting (line 187):\
> **A7**: When a positive class is OOD for a KAN trained on samples of a negative class, the InD score will be low for that KAN. If another KAN is trained on the positive class, the maximization procedure will flag this sample as InD, as this second KAN will return a high InD score. If the negative class is actually OOD, none of the KANs will return a high InD score, and the maximization procedure will correctly flag the sample as OOD. We clarified this relationship with the InD score at lines 196-197 in the revised manuscript.
>
> **Q8**: Robustness for NaN-Enriched Data:\
> **A8**: We thank the reviewer for raising this interesting point. With modifications to the KAN grid, it should be possible to handle NaN values. We hypothesize that assigning an individual spline coefficient to handle NaN values should suffice. However, since there is no distance relation between NaN and other spline coefficients, the smoothing operations of splines around NaN will be affected.
>
> We sincerely appreciate your detailed feedback. Your suggestions have been very valuable in refining our paper, and we believe that the added experiments and improvements in clarity enhanced the overall contribution of our work.

---

> > ### Comment · Reviewer_Asxo · 2024-11-26
> >
> > I appreciate the new experiments but the scalability to large datasets still remains an issue (time constraints indeed are harsh for the rebuttal). This appears to be a very borderline case among all reviewers because of this trait. So, I will keep my rating but will increase my 'soundness' score to acknowledge authors' explanations.

---

> > > ### Author Response · Authors · 2024-11-30
> > >
> > > As you suggested, we conducted additional experiments on the OpenOOD ImageNet-1K (full-spectrum) benchmark.
> > > Our method achieves first place also on this leaderboard, outperforming the previously best method by 2\%.
> > > We include a detailed table of these results in a general comment for your reference.
> > >
> > > We hope this addresses your remaining concerns.
> > > Thank you once again for your valuable feedback and for considering our additional experiments.

---

### Author Response · Authors · 2024-11-25
**Response to reviewer feedback**

We thank all the reviewers for their valuable feedback, which has significantly improved the quality of our work.

Detailed answers and additional experiments regarding all the concerns raised by the reviewers can be found in the individual replies of each review.

A significant concern raised by all reviewers regards the scalability of our method.
To address this, we conducted an extensive experiment on the large-scale ImageNet-200 full-spectrum benchmark.
This benchmark is particularly challenging as it includes five times more samples with images that are seven times larger compared to the CIFAR datasets.
Additionally, the full-spectrum version increases the detection challenge by enriching the InD test set with extra covariate-shifted samples.

Our results, illustrated in Table 2 of Section 3.2 of the revised manuscript, show that our method ranks first, surpassing the previous best approach (ASH) by approximately 4\% on the overall average AUROC metric.
These results demonstrate the scalability and effectiveness of our method in handling large-scale datasets and complex real-world scenarios.

We also expanded Appendix A.11 with a detailed discussion on the method's complexity.
Our analysis indicates that the most impactful factor is the dataset size, and our method exhibits a similar scaling law to other approaches.
This detailed discussion provides insights into the computational efficiency and practicality of our approach, reinforcing its applicability to large-scale problems.

We are grateful to the reviewers for their suggestion to include this experiment, as it has enhanced the robustness and comprehensiveness of our paper.

---

> ### Author Response · Authors · 2024-11-30
> **Evaluation on ImageNet-1k**
>
> We further evaluated our method on the challenging ImageNet-1k (full-spectrum) benchmark.\
> Remarkably, our method ranks first, outperforming the previous best approach (NAC) by approximately 2% in the overall average AUROC metric.
> Detailed results can be found in the table below.
>
> | **Method** | **SSB-hard** | **NINCO** | **iNaturalist** | **Textures** | **OpenImage-O** | **Avg Near** | **Avg Far** | **Avg Overall** |
> |------------|--------------|-----------|-----------------|--------------|-----------------|--------------|-------------|-----------------|
> | OpenMax    | 53.79        | 60.28     | 80.30           | 73.54        | 71.88           | 57.03        | 75.24       | 67.96           |
> | ODIN       | 54.22        | 60.59     | 77.43           | 76.04        | 73.40           | 57.41        | 75.62       | 68.34           |
> | MDS        | 39.22        | 52.83     | 54.06           | 86.26        | 60.75           | 46.02        | 67.02       | 58.62           |
> | MDSEns     | 37.13        | 47.80     | 53.32           | 73.39        | 53.24           | 42.47        | 59.98       | 52.98           |
> | RMDS       | 56.61        | 67.50     | 73.48           | 74.25        | 72.13           | 62.06        | 73.29       | 68.79           |
> | Gram       | 51.93        | 60.63     | 71.36           | 84.83        | 69.40           | 56.28        | 75.20       | 67.63           |
> | ReAct      | 55.34        | 64.51     | 87.93           | 81.08        | 79.34           | 59.93        | 82.78       | 73.64           |
> | VIM        | 45.88        | 59.12     | 72.22           | 93.09        | 75.01           | 52.50        | 80.10       | 69.06           |
> | KNN        | 43.78        | 59.86     | 67.79           | 90.29        | 69.98           | 51.82        | 76.02       | 66.34           |
> | ASH        | 54.66        | 66.38     | 89.23           | 89.53        | 81.47           | 60.52        | 86.75       | 76.25           |
> | SHE        | **58.15**        | 64.27     | 84.71           | 87.48        | 76.92           | 61.21        | 83.04       | 74.31           |
> | GEN        | 52.95        | 62.73     | 78.47           | 71.82        | 72.62           | 57.84        | 74.31       | 67.72           |
> | NAC        | 52.48        | 66.49     | 88.92           | 92.77        | 80.76           | 59.48        | 87.48       | 76.28           |
> | **KAN**        | 55.88        | **69.55**     | **91.55**           | **93.45**        | **82.15**           | **62.71**        | **89.05**       | **78.52**           |
>
> This benchmark includes over 1.2 million samples and utilizes a larger backbone network (ResNet-50).
> The results further validate our method's capability to handle complex, large-scale problems.\
> We currently cannot update our manuscript, but if given the opportunity, we will include these results in the camera-ready version of our paper.

---

> > ### Author Response · Authors · 2024-12-02
> > **Training Time**
> >
> > We would like to provide additional details regarding the ImageNet-1K experiment reported in our previous comment.
> > For this experiment, we employed class-based partitioning, resulting in 1000 clusters.
> > However, we reduced the number of outputs for each model from 1000 to 10 classes by randomly grouping labels together.
> > This adjustment is motivated by the fact that with 1000 clusters, the problem tackled by each model is greatly reduced, and thus the model's capacity can also be reduced.
> >
> > As a result, the training time per sample of our model is slightly lower than that for the ImageNet-200 benchmark: approximately 1.6ms per sample compared to 1.9ms per sample for ImageNet-200.
> > This indicates that our method remains efficient and robust even with a large number of clusters.
> > We hope this clarification alleviates any concerns regarding the scalability and efficiency of our approach.

---

### Meta-Review · Area_Chair_haj7 · 2024-12-21

**Metareview:**

This paper presents an OOD detection method using properties of the KAN model. They compare activation patterns of a trained KAN against an untrained one and look for patterns that will demonstrate OODness based on the properties of the KAN. They show experiments on benchmark datasets demonstrating their superior OOD detection performance.

Strengths: Interesting and novel idea, positive results on benchmarks
Weaknesses: Limited evaluation on real, larger scale datasets, performance gains are modest

More evaluations on larger scale datasets would make the paper's contributions stronger and lend more credence to the detection method. The performance gains seem to reduce as the experiments move to large scale (unclear what the statistical difference is from the new table for Imagenet - 1k)  - more complex/large datasets will help increase understanding of this as well.

The paper remains in borderline with all reviewers after the new experiments as well - this is understandable given the short rebuttal time, but provides opportunities for further improvement of the paper.
If accepted, the authors will need to include the new large scale experiments with clear discussion on the computational aspects as well as include statistical estimates in the table to judge how statistically significant the 2% performance increase is.

**Additional Comments On Reviewer Discussion:**

The main concern amongst reviewers was in applicability of the method beyond simple datasets. Due to this, there were no clear champions for the paper.
The new experiment on ImageNet-1k is a large scale experiment that still demonstrates the performance of the method but I did not see the same format of results as in the paper (metric and variance in metrics to judge if the results are statistically significant). Reviewer Asxo judges that the authors probably rushed to submit this and the ask for more evaluations on challenging datasets, while very important, is a substantial ask in the short rebuttal period - it is unclear how the new results should be assessed.

The reviewers raised other points as well:
* Choice of k-means for clustering
* Performance in regression tasks
* Computational costs
* Other clarifications regarding training process, influence of model capacity, amongst others
I feel the authors have answered these questions in the rebuttal though most reviewers were unresponsive. Reviewer Asxo feels the paper is borderline accept after all the rebuttals.

---

### Decision · Program_Chairs · 2025-01-22

Accept (Poster)